# Majorana zero modes in impurity-assisted vortex of LiFeAs superconductor

Lingyuan Kong [1,6], Lu Cao [1,2,6], Shiyu Zhu [1,6], Michał Papaj [3,6], Guangyang Dai[1,2], Geng Li [1,2], Peng Fan[1,2], Wenyao Liu [1,2], Fazhi Yang[1,2], Xiancheng Wang[1], Shixuan Du[1,2,4], Changqing Jin [1,2,5], Liang Fu[3], Hong-Jun Gao [1,2,4 ✉] & Hong Ding [1,4,5 ✉]

The iron-based superconductor is emerging as a promising platform for Majorana zero mode, which can be used to implement topological quantum computation. One of the most significant advances of this platform is the appearance of large vortex level spacing that strongly protects Majorana zero mode from other low-lying quasiparticles. Despite the advantages in the context of physics research, the inhomogeneity of various aspects hampers the practical construction of topological qubits in the compounds studied so far. Here we show that the stoichiometric superconductor LiFeAs is a good candidate to overcome this obstacle. By using scanning tunneling microscopy, we discover that the Majorana zero modes, which are absent on the natural clean surface, can appear in vortices influenced by native impurities. Our detailed analysis reveals a new mechanism for the emergence of those Majorana zero modes, i.e. native tuning of bulk Dirac fermions. The discovery of Majorana zero modes in this homogeneous material, with a promise of tunability, offers an ideal material platform for manipulating and braiding Majorana zero modes, pushing one step forward towards topological quantum computation.

---

[1] Beijing National Laboratory for Condensed Matter Physics and Institute of Physics, Chinese Academy of Sciences, Beijing, China. [2] School of Physical Sciences, University of Chinese Academy of Sciences, Beijing, China. [3] Department of Physics, Massachusetts Institute of Technology, Cambridge, Massachusetts, USA. [4] CAS Center for Excellence in Topological Quantum Computation, University of Chinese Academy of Sciences, Beijing, China. [5] Songshan Lake Materials Laboratory, Dongguan, Guangdong, China. [6] These authors contributed equally: Lingyuan Kong, Lu Cao, Shiyu Zhu, Michał Papaj. ✉email: hjgao@iphy.ac.cn; dingh@iphy.ac.cn

The recent realization of pristine Majorana zero modes (MZMs) in vortices of iron-based superconductors (FeSCs)[1–5] provides a promising platform for long-sought-after fault-tolerant quantum computation[6–15]. A large topological gap between the MZMs and the lowest excitations enabled detailed characterization of vortex MZMs in those materials[16–23]. Despite those achievements, a practical implementation of topological quantum computation based on MZM braiding[2,24] remains elusive in this new Majorana platform. Among the most pressing issues is the inhomogeneity of the existing FeSC Majorana materials that destroys MZMs during the braiding process[25]. Thus, the realization of vortex MZMs in a truly homogeneous material of stoichiometric composition and with a charge neutral cleavage surface is highly desirable.

LiFeAs, which belongs to the family of iron pnictides (Fig. 1a)[26–30], has multiple topological bands[31] owing to a similar $p$-$d$ inversion mechanism as its iron chalcogenide cousins. A recent angle-resolved photoemission spectroscopy (ARPES) experiment[31] reported two Dirac cones close to the Fermi level, which are the surface Dirac fermion of a topological insulator (TI) phase and the bulk Dirac fermion of a topological Dirac semimetal (TDS) phase, respectively (Fig. 1b). Although both of them can lead to Majorana quasiparticle in a vortex, the exact type of the realized excitation, either the localized single zero modes or mobile helical modes are highly dependent on the nature of the underlying bands incorporated in the vortex quasiparticle excitations[32,33]. Thus altering the chemical potential ($\mu$)

in LiFeAs is expected to change the topological phase of the material and tune the type of Majorana quasiparticles accordingly. The rich topological band structure near the Fermi level of LiFeAs superconductor makes it a fertile playground for studying Majorana physics in both TI and TDS phases. More importantly, LiFeAs has remarkably homogeneous electronic properties[29,34] (Supplementary Fig. 2b, c), owing to its dopants-free stoichiometric bulk and a charge neutral cleavage surface in between the lithium double layers (Fig. 1a). It can be found that majority of the area on the as-cleaved surface in LiFeAs is clean and uniform with an ordered square lattice (Fig. 1c). Sporadically, some native impurities form spontaneously, likely to reduce the surface energy and are distributed sparsely over the intrinsically homogenous background[35–37]. That offers a great promise for creating, manipulating, and tuning MZMs in this homogeneous material. Although the rich topological band structure and the homogeneity make LiFeAs a seemingly better FeSC Majorana material, previous scanning tunneling microscopy/spectroscopy (STM/S) measurements showed the absence of vortex MZM[27–30], which constituted a major puzzle in the field of vortex-based Majorana platforms[31].

In this work, we demonstrate experimentally that the stoichiometric superconductor LiFeAs[26] is indeed a good candidate to overcome the obstacle of inhomogeneity suffered by the previous FeSC Majorana materials. Using scanning tunneling microscopy, we discover that the MZMs, which are absent on the natural surface[27–30], can appear in vortices influenced by native

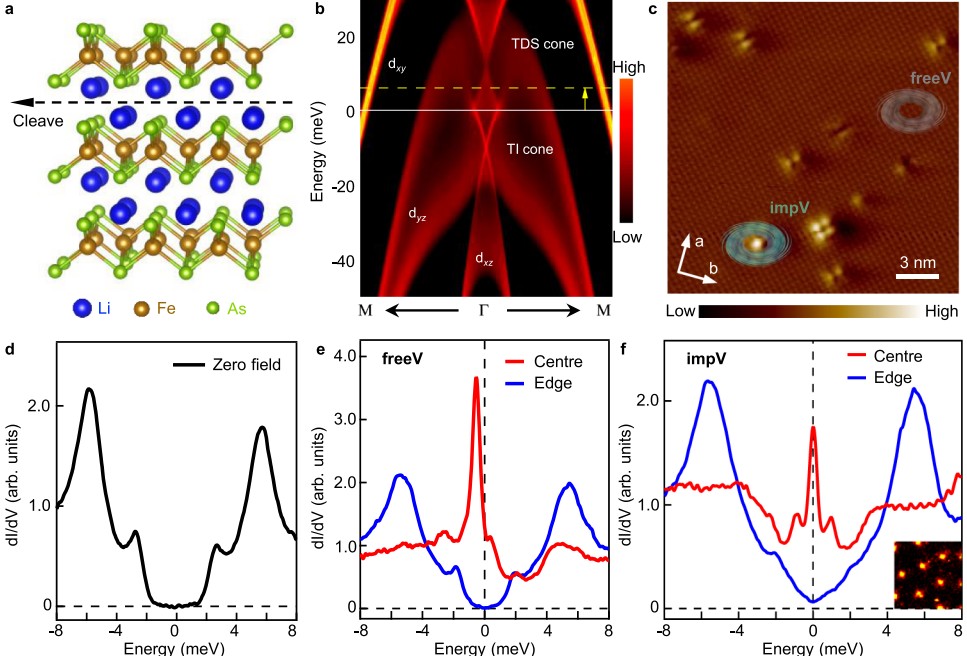

**Fig. 1 Vortices with and without zero-bias conductance peaks in LiFeAs. a** Crystal structure of stoichiometric LiFeAs. The natural cleavage plane is indicated by the black dashed line. **b** (001)-surface-projected band structure of LiFeAs. [Adopted from Fig. 2d of Zhang et al.[31], the chemical potential is adjusted to match the experimental value as shown in supplementary of Zhang et al.[31] (white-solid line), the yellow dashed line and arrow indicate the chemical potential shift around impurities as observed below.]. Two topological cone-like bands appear in the calculation near the Fermi level. The lower one is the Dirac surface state of a topological insulator (TI) phase, and the upper one is the bulk Dirac fermion of a topological Dirac semimetal (TDS) phase. **c** Atomic resolution STM topography of LiFeAs (scanning area, 20 nm by 20 nm), axes a and b indicate the Fe–Fe bond directions. The native impurities are sparsely distributed on the homogeneous surface which maintains large, well-ordered surface area with its electronic properties unchanged (Supplementary Fig. 2b, c). The vortices on LiFeAs surface belong to two classes: free vortices (freeV) on the clean surface (gray symbol) and impurity-assisted vortices (impV) pinned to native impurities (green symbol). **d** A typical tunneling conductance spectrum measured on a clean area of LiFeAs surface under zero field. Two bulk superconducting gaps are identified: $\Delta_1 = 2.7$ meV (due to the outer hole Fermi surface), and $\Delta_2 = 5.8$ meV (due to the inner hole Fermi surface). **e** Sharp non-zero-energy vortex bound states at the center of a freeV. **f** Zero-energy vortex bound states, accompanied by a pair of energy-symmetric non-zero vortex bound states in a typical impV. Inset: the vortex lattice on a zero-bias conductance (ZBC) map under 2.0 T (scanning area, 100 nm by 100 nm). The settings are: sample bias $V_b = -5$ mV; tunneling current $I_t = 200$ pA; and temperature $T_{exp} = 400$ mK.

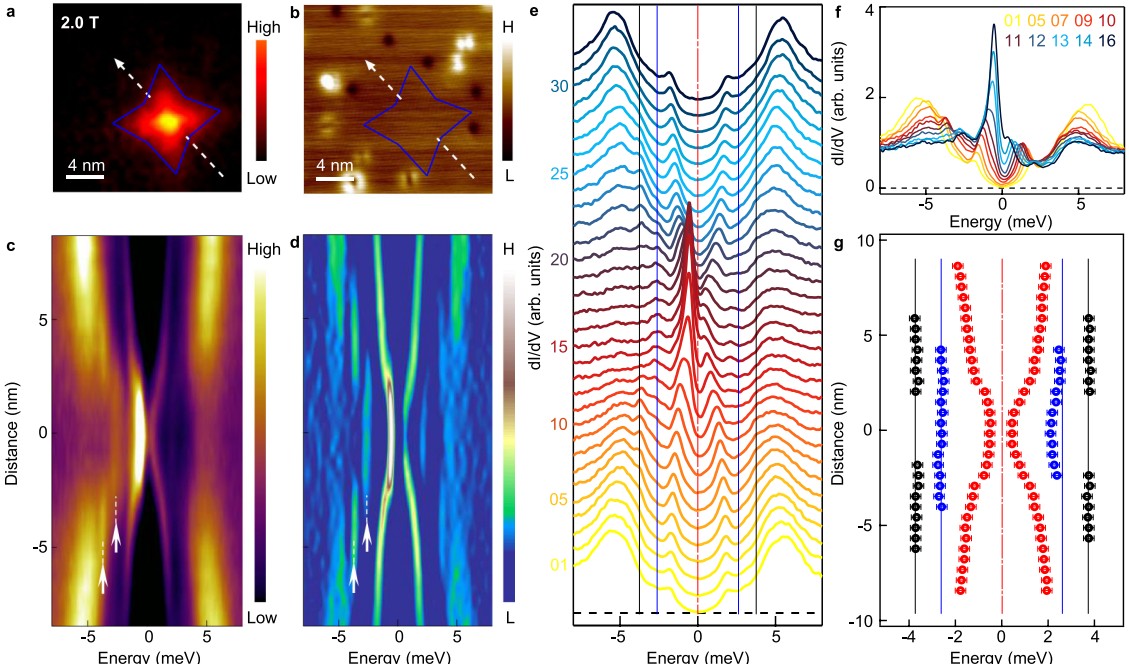

**Fig. 2 Coexistence of discrete and dispersive vortex bound states in a free vortex. a** A ZBC map around a freeV. The zero-energy LDOS exhibits star-like shape, with its quasiparticle tails along the nearest As–As directions. **b** The corresponding topography of **a**. The "L" and "H" in the label of the color bar stand for "low" and "high" respectively. This abbreviation is also used in other figures of this work. **c** Line-cut of $dI/dV$ measured in the freeV along the white dashed lines indicated in **a** and **b**. **d** Curvature intensity plot of **c**. The white dashed lines with arrows indicate the two discrete vortex bound states at high energy. **e** Waterfall-like plot of **c**. **f** Plot of several overlapping spectra selected from **e**. **g** Extracted energy positions of the vortex bound states in **c–e**. Two sets of vortex bound states could be identified, i.e. $d_{yz}$ orbital related discrete levels (blue and black symbols), and $d_{xy}$ orbital related dispersive levels (red symbols). The error bars are estimated by the instrumental resolution.

impurities. Our detailed analysis and model calculations explain the emergence of those vortex MZMs owing to native tuning of the bulk Dirac fermions, paving a way towards tuning MZMs by controllable methods such as impurity planting, mechanical stress, and electrostatic gating.

## Results

To introduce a tuning parameter necessary in this largely homogeneous system, we took advantage of sparsely distributed impurities formed on the cleaved sample[35–37] as means to enable the emergence of the MZMs. To investigate the morphology of vortex bound states experimentally, we performed low-temperature ($T_{exp} = 400$ mK), high-resolution (better than 0.28 meV) STM/S measurements on the surface of LiFeAs superconductor. The bare surface after in situ cleavage is formed by lithium atoms. We observe atomically resolved square lattice of lithium atoms[38] in a sizable region free of impurities (Fig. 1c). Two superconducting gaps ($\Delta_1 = 2.7$ meV; $\Delta_2 = 5.8$ meV) are clearly observed under zero field (Fig. 1d), consistent with the previous STM/S results[27,30] (see detailed gap assignment in Methods section). Besides the clean region, some spontaneously formed native impurities are also observed on the cleaved surface (Fig. 1c). With a 2 T magnetic field applied perpendicular to the sample surface, we find that the vortices not only appear as free vortices (freeVs) in the clean regions, but also appear as impurity-assisted vortices (impVs) pinned to the positions of the sparsely distributed impurities (see a schematic illustration in Fig. 1c). Besides the ordinary non-zero-energy peaks observed in all freeVs (Fig. 1e), a pronounced zero-bias conductance peak (ZBCP) emerges in some impVs (Fig. 1f). In addition, the ZBCP is usually accompanied by a pair of energy-symmetric side peaks, which are located at about ±0.9 meV in Fig. 1f, while the spectrum recovers the superconducting gap feature at the impV edges

(Supplementary Fig. 1a, b). These phenomena are strikingly similar to the MZM and accompanied integer-quantized vortex bound states observed in the two previously identified FeSC Majorana materials, Fe(Te,Se)[17] and CaKFe$_4$As$_4$[22].

**Finite energy modes in a free vortex.** We first discuss the typical behavior of vortex bound states of freeVs in which no ZBCP are observed. We show detailed line-cut measurements across a freeV in Fig. 2. Similar to what has been observed previously[27,28], the most pronounced spectral features are two dispersive side peaks (Fig. 2c) at non-zero energies (Fig. 2e–g). Owing to the high resolution of our data, we further distinguish two additional discrete vortex bound states located at ±2.6 meV and ±3.7 meV, respectively, which display the quantized behavior[39]. Those discrete levels can be observed directly in the raw data (Fig. 2c) and be recognized more clearly in a curvature plot[40] (Fig. 2d). The extraction of the spatial evolution of those vortex bound states is shown in Fig. 2g. By a more detailed analysis (see in the Methods section), we find that the dispersive and discrete vortex bound states can be attributed to the outmost $d_{xy}$ band and the inner $d_{yz}$ bulk band, respectively. This implies that the topological bands are likely decoupled from the quasiparticle excitations in a freeV.

**Zero energy modes in an impurity-assisted vortex.** To characterize the ZBCPs, we next focus on an impV (Fig. 3a). Through a line-cut measurement across the impV (Fig. 3c), we find that the ZBCPs (Fig. 1f) remain fixed at zero energy (Supplementary Fig. 1a), with their intensity gradually decreasing to zero when moving away from the vortex center. This non-splitting behavior can be also observed clearly from a waterfall plot (Fig. 3e) and an overlapping plot (Fig. 3f) of the same impV. This phenomenon is similar to the behavior reported for the MZMs in the known FeSC Majorana materials[2]. However, unlike the nearly isotropic

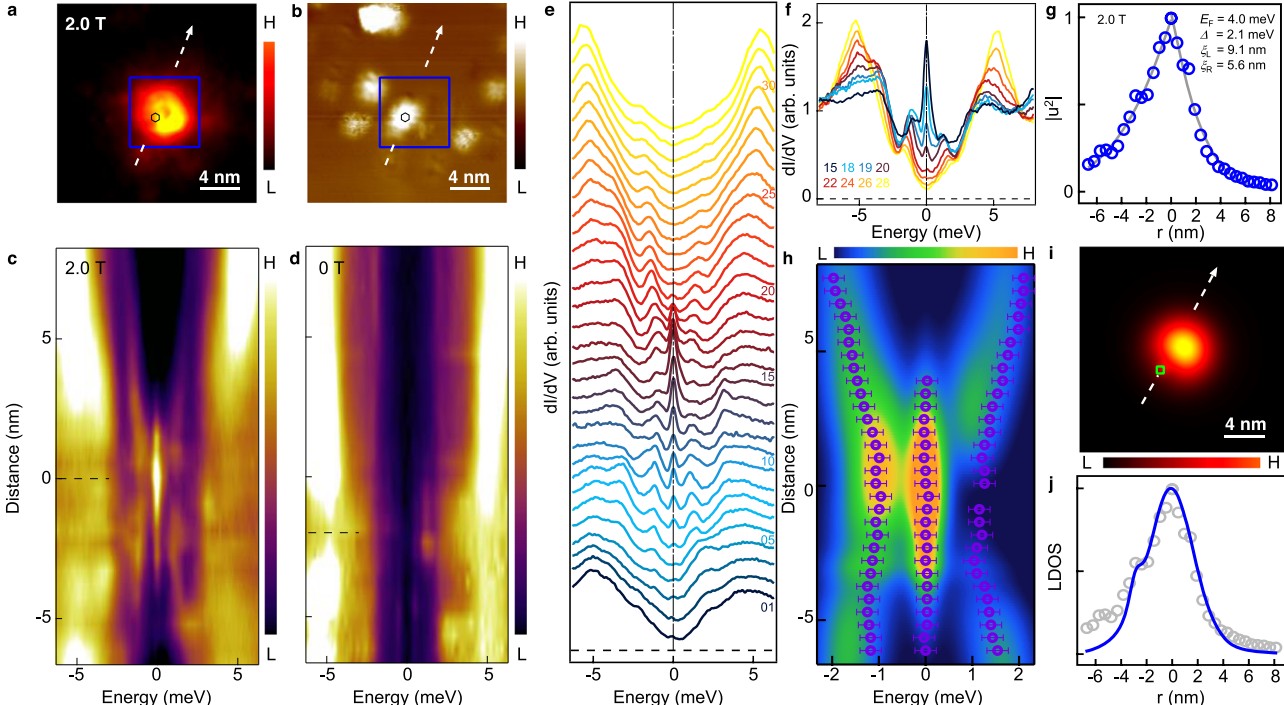

**Fig. 3 Asymmetric Majorana zero modes in an impurity-assisted vortex. a** A ZBC map of an impV. **b** Corresponding STM topography of **a**. Blue boxes mark the same area where the vortex appears, while the black hexagons mark the center of the impurity. **c** Line-cut of d$I$/d$V$ measured under 2.0 T along the white dashed line indicated in **a**, which demonstrates the spatial evolution of the vortex bound states. **d** Line-cut of d$I$/d$V$ measured under 0 T and at the same location as in **c**, which demonstrates the spatial evolution of impurity bound states. The black dashed lines in **c** and **d** indicate the center positions of the vortex and the impurity, respectively. **e** Waterfall-like plot of **c**. **f** Eight spectra selected from the 2.0 T data in **e**. **g** Blue symbols: zero-bias d$I$/d$V$ line profile of zero-bias conductance peaks (ZBCPs) measured along the dashed line in **a**. The parameters of underlying topological bands (i.e. $\Delta = 2.1$ meV; $E_F = 4.0$ meV) are extracted by fitting the data with an analytical Majorana wavefucntion[2,11,22,43] (gray curves). While the parameters of $\Delta$ and $E_F$ used in the fitting of left and right side are same, the fitting parameter of wavefunction decay length of the two sides are different, i.e. $\xi_L = 9.1$ nm; $\xi_R = 5.6$ nm. **h** Simulated local density of states line-cut across a vortex with impurity. The purple symbols are the measured energies of vortex bound states shown in **c** and **e**. The error bars are estimated by the instrumental resolution. **i, j** Simulation of a Majorana wavefunction influenced by an impurity, a two dimensional zero-energy local density of states and its line profile (blue curve) are shown, respectively. The green square in **i** indicates the impurity center. **j** is traced along the white line indicated in **i**. The gray symbols in **j** are the experimental results shown in **g**. The parameters used in simulation of **h–j** are based on the fitting results from **g**: $\Delta = 2.1$ meV; $E_F = 4.0$ meV, $\xi_0 = 3.6$ nm. The energy broadening used in the simulation of **h** are 0.5 meV for ZBCPs and 1 meV for other vortex bound states, as measured in this system (Supplementary Fig. 1a, b).

MZMs discussed previously[2,22], the intensity of ZBCPs across the impV shows considerable asymmetry (Fig. 3g). This asymmetry is intimately connected to the influence of the impurity potential. As shown in Fig. 3a, b, the intensity maximum of vortex bound state is shifted from the center of impurity. This can be recognized more clearly by comparing the position of the strongest ZBCP in the vortex line-cut and the position of the weakest superconducting coherence peak in the zero field line-cut (Supplementary Fig. 1e). We marked those two positions as the horizontal dashed bars in Fig. 3c, d, respectively. The impurities may disturb the density of Cooper pairs asymmetrically with respect to the center of the vortex core, and lead to an anisotropic vortex. Beside the isolated ZBCPs, two energy-symmetric side peaks display the dispersive behavior across the vortex core. By fitting the d$I$/d$V$ spectrum measured at the vortex center, we find that the full width at half maximum (FWHM) of ZBCP is ~0.5 meV and that of the side peaks is larger than 1 meV (Supplementary Fig. 1a, b). In the spectra of quantized topological vortices near the zero-doping-limit[17,41,42] (when Fermi level is located at the Dirac point), a large quasiparticle gap develops between MZM and the lowest excitations. At the same time, the energy separations between the subsequent higher levels are much smaller owing to superconducting quantum confinement. Due to the increased energy broadening of non-zero-energy vortex bound

states the high energy levels merge into dispersive peaks, while the zero mode still remains isolated. Therefore, we conclude that the side peaks visible in our measurements are built of multiple independent vortex bound states that are overlapping due to energy broadening.

**Fit to asymmetric Majorana zero mode**. To further substantiate our interpretation of the nature of the higher energy dispersive vortex bound state and explain the asymmetry of the zero-energy state, we perform theoretical calculations using a lattice model (see details in Methods section). The model is based on a 2D proximitized Dirac fermion with a superconducting vortex and an impurity placed off the vortex center. The parameters used in the calculations are based on fitting to the analytical model of Majorana wavefunction[2,11,22,43] (Fig. 3g). With broadening modeled using an effective temperature resulting in FWHM comparable to the measured value (Supplementary Fig. 1a, b), a good agreement with the experimental data is obtained (Fig. 3h). As the chemical potential $\mu \approx 1.9\Delta$, the first non-zero energy state lies at $E_1 \approx 0.5\Delta$ and the following states are closely spaced in energies. Their spacing is smaller than the peak broadening and thus they appear to be a single dispersive state (a corresponding simulation without broadening applied see below in the last

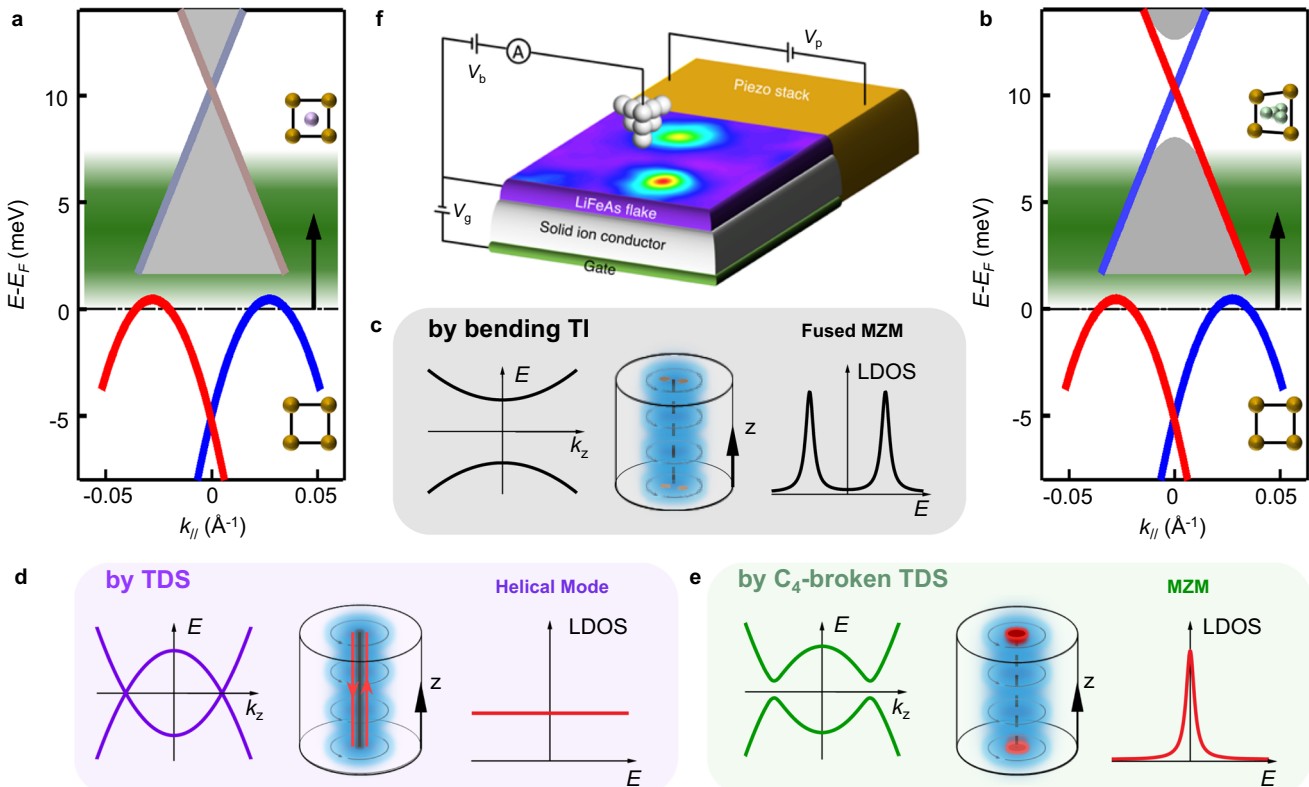

**Fig. 4 Stabilizing Majorana zero modes by tuning bulk Dirac fermion. a**, **b** Schematic depiction of topological band structure of LiFeAs influenced by impurities. Band dispersion of the lower Dirac surface state and the upper bulk Dirac fermion is extracted from ARPES measurements[31]. The Fermi level in the diagram follows from the measured chemical potential ($\mu$) of LiFeAs. The energy separation between the two Dirac points is ~15 meV, the Dirac point of topological surface states is located at ~−5 meV. The green shaded region in **a** and **b** indicates the available $\mu$ range tunable by impurities as demonstrated in Figs. 5 and 6 and Supplementary Fig. 2. Stronger impurity not only tends to induce a larger electron doping effect (Fig. 7a), but also induces a larger asymmetric stress which causes lattice distortion of the bulk and breaks the $C_4$ symmetry of the square lattice near the impurity (inset of Fig. 4b). Thus, the upper bulk Dirac cone remains intact in the region influenced by weaker impurities (**a**), but becomes gapped in the region near the stronger impurities (**b**). **c**–**e** The dispersion of the lowest vortex bounds states in the bulk along $k_z$ direction (left panel), the diagram of the vortex line (middle panel), and the local density of states (LDOS) of Majorana modes (right panel) in the cases of the bending topological insulator (TI) surface states (near the Fermi level in **a** and **b**), topological Dirac semimetal (TDS) bands which maintains full symmetries (green shaded region in **a**), and $C_4$-symmetry broken TDS bands (green shaded region in **b**), respectively. Majorana modes appear as: fused fermionic bound state, two pairs of mobile helical Majorana modes in the nodal vortex line, and single Majorana zero modes (MZM), respectively. **f** An experimental design to realize a fast, tunable Majorana device using LiFeAs vortex. By combining efficient gating effect (e.g. field-effect transistor using solid ion conductors as gate dielectric[47]) and uniaxial deformation (e.g. a device controlled by piezo stack which is tightly connected to the sample[48,49]), the phase transition among trivial vortex bound states, helical Majorana modes and localized MZM could be detected continuously by an in situ STM measurements in the future.

figure). Moreover, the presence of a strong impurity introduces asymmetry in the zero-energy state wavefunction (Fig. 3i) consistent with the measurements, significantly increasing the decay length in that direction (the left side in Fig. 3j), while the opposite side remains largely compatible with the analytical solution (the right side in Fig. 3j). The good consistency between the experiment and theoretical simulations strongly support that the observed ZBCP is an asymmetric MZM from the Dirac states near the zero-doping limit[17,41,42].

**Majorana mechanism in LiFeAs**. We now demonstrate a possible mechanism for the distinct behavior of different types of vortices and the tunability of Majorana modes in LiFeAs. First, we analyze the freeV cases. As shown in Fig. 4a, b, which depict the topological bands of LiFeAs obtained from a previous ARPES measurement[31], in clean regions, $\mu$ is located within the Dirac surface states of the TI phase. The puzzle of absence of MZMs in previous measurements of freeV[27–30] can now be resolved by noticing the behavior of Dirac surface states observed in the

ARPES work[31]. As the dispersion bends back, it tends to form a Rashba-like dispersion in the upper branch of Dirac surface state (Fig. 4a, b), which leads to the Fermi level crossing the helical Dirac electrons twice. Accordingly, two MZMs emerge in a core of freeV (the middle panel of Fig. 4c). The pairs of unprotected MZMs can fuse with each other immediately to become fermionic bound states at the gap edge (the right panel of Fig. 4c). Consequently, the vortex bound states in freeVs have no zero modes with their behavior fully explained by the bulk bands (Fig. 2g). It is worth noting that the TI phase can also support single vortex MZM in LiFeAs, if the hole doping can be introduced in the sample that moves the Fermi level to the lower portions of the Dirac cone where the Fermi level crosses the surface state only once.

For the impV cases, we propose that the presence of an impurity can significantly affect its vicinity in several ways. First, the impurity can provide electron doping, lifting $\mu$ above the TI regime. This conjecture is supported experimentally, as demonstrated in the Methods section (Figs. 5 and 6 and Supplementary Fig. 2). As shown in Fig. 4a, b, the energy separation between the

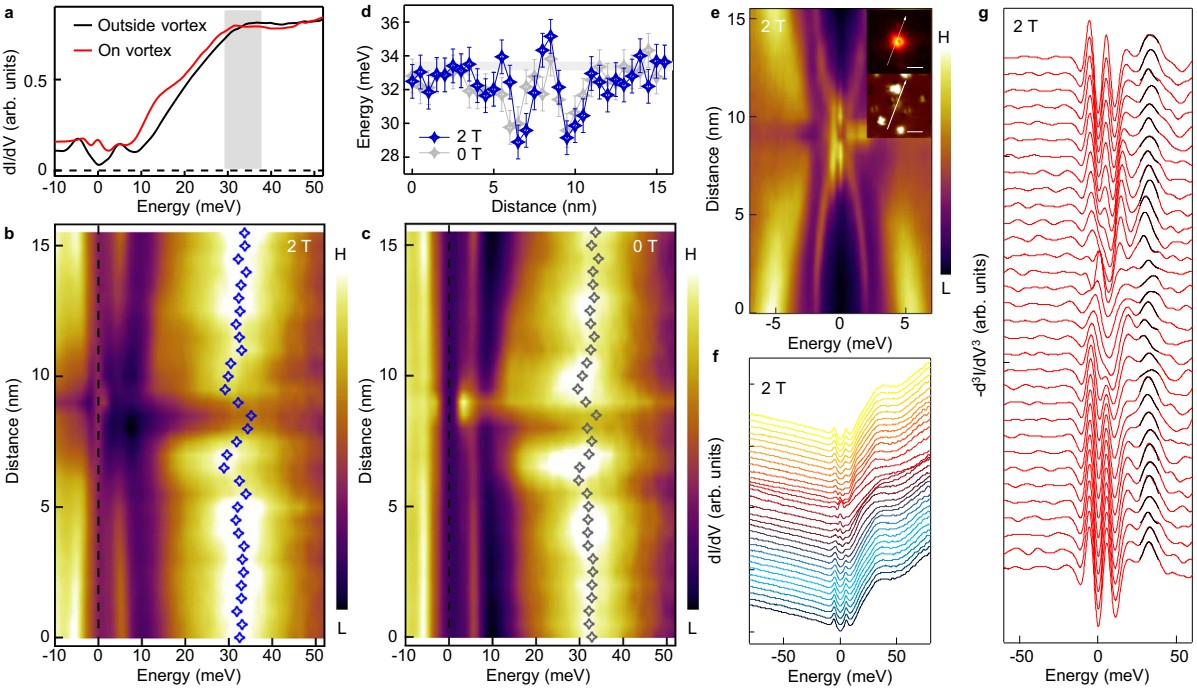

**Fig. 5 Evidence of local electron doping around an impurity-assisted vortex. a** Wide range d$I$/d$V$ spectra measured at an impV (red curve) and on a clean surface region without impurities (black curve). The gray shaded region emphasizes the shift of the $d_{xy}$ band top, indicating electron doping around the impV. **b** Wide range line-cut intensity plot with a linear background subtraction for an impV (see **e**, for basic information). The blue symbols are extracted energies of the band top by the method introduced in **f** and **g**. **c** Same as **b**, but measured under zero field. **d** Comparison of spatial evolution of the band top energies across the impV measured under 2.0 T and 0 T. The gray thick line indicates the reference band top energy measured on a clean area without impurities (Supplementary Fig. 2). The error bars are around 1 meV, determined by the maximum spatial variation of the reference energy ($E_{BT}$ = 33.4 meV) measured in Supplementary Fig. 2a. **e** Short range line-cut intensity plot of the impV. Inset: corresponding ZBC map and topography (scale bar: 5 nm). **f, g** Numerical method for the $\mu$-shift extraction. **f** Raw data of the wide range scan shown in **b**. The $\mu$-shift across the impV is visible in the spectra overlapping plot. **g** Negative second derivative of the spectra shown in **f**. The signal of the peak in d$I$/d$V$ spectra is enhanced as a peak in the negative second derivative spectra. In order to establish the $\mu$-shift, we extract the energy of the $d_{xy}$ band top by a simple Gaussian fit. The $\mu$-shift in this impV is determined to be 4.5 ± 1 meV.

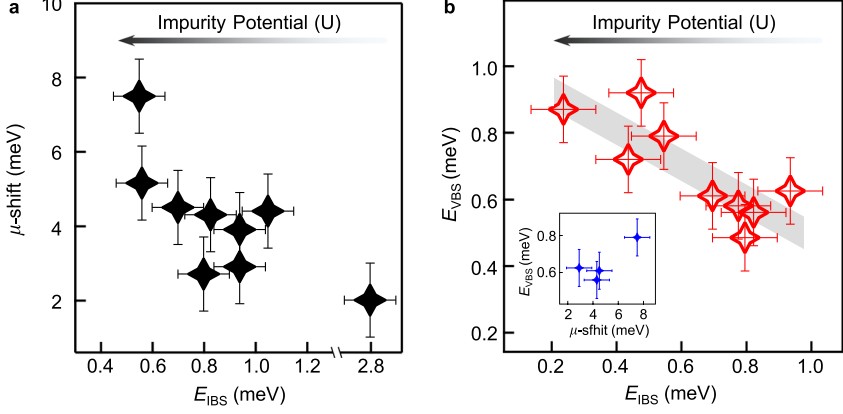

**Fig. 6 Relationship between vortex tunability and impurity potential. a** Summary of the relationship between the chemical potential shift ($\mu$-shift) and the energy of the lowest impurity bound states ($E_{IBS}$) among 9 impVs (some impVs collected in **a** with weak impurities do not have MZMs, see individual behaviors of each impV in Supplementary Tables 1 and 2). $\mu$-shift is extracted as the maximum band top energy deviation at the impVs from the reference energy ($E_{BT}$). $E_{IBS}$ is extracted from line-cut intensity measurements under zero field, and along the same line as the measurements under 2.0 T in each vortex. **b** Summary of the relationship between the energy of the lowest non-zero energy vortex bound states ($E_{VBS}$) and the $E_{IBS}$ among 9 impVs which have MZMs (Supplementary Table 1). $E_{VBS}$ is also the topological gap that determines the protection of MZM. Inset: a summary of the relationship between $E_{VBS}$ and $\mu$-shift among 4 impVs for which both quantities are measured. A smaller $E_{IBS}$ indicates a stronger impurity potential. The behavior shown in **a** and **b** indicates that vortices pinned to stronger impurities simultaneously have larger $\mu$-shift and larger $E_{VBS}$. It implies that impurity-induced local electron doping shifts $\mu$ towards the TDS point, which is further supported by the data shown in the inset of **b**. The error bars of $E_{VBS}$ and $E_{IBS}$ are 0.1 meV. The error bars of $\mu$-shift are defined as same as that in Fig. 5d.

two Dirac points is only ~15 meV[31], so that the electron doping can easily push the region surrounding the impV into the upper TDS regime. It has been predicted that within the TDS phase two pairs of helical Majorana modes propagating along the vortex line will emerge (the middle panel of Fig. 4d). Such a nodal vortex line would exhibit a constant local density of states[32,33] (the right panel of Fig. 4d). Furthermore, a recent theory proposed that the nodal vortex line could be gapped out by a $C_4$ symmetry breaking perturbation[33] (the left panel of Fig. 4e). In this way, a single MZM could be stabilized on the sample surface (the middle and the right panel of Fig. 4e). Remarkably, we observe that the impurities of the impV with ZBCPs have an asymmetric shape which enhances the asymmetric stress, consequently breaking the $C_4$ symmetry adequately[44,45] (inset in Fig. 4b). This effect makes the nodal vortex line to be full-gapped and transforms the helical Majorana modes into a well localized MZM in impVs. Finally, we note that the impurity must affect sufficiently large surrounding volume, so that the topological vortex line extends into the bulk. This is necessary to allow for adequate spacing between its two ends and in consequence prevents the hybridization of the two emerging MZM located there. This requirement for strong impurity influence is further corroborated by the observation that MZMs are absent in most impVs pinned to weak impurities (see details in the Methods section). In this case, the characteristics of vortex bound states are either similar to the freeV (Fig. 4c) or the case of nodal vortex line that emerged from full-symmetric TDS phase (Fig. 4d). More detailed characteristics of strong impurities, which can create vortex MZM in impVs, and weak impurities, which can not create vortex MZM in impVs, are shown in Supplementary Figs. 3–4, Supplementary Tables 1–2, and the related content in the Methods section.

## Discussion

Our work demonstrates the first realization of MZMs in the homogeneous LiFeAs superconductor. Besides revealing new Majorana physics in bulk Dirac fermions, which solves a major puzzle in the field of vortex-based Majorana modes, it may also bridge the gap between quantum physics and quantum engineering in FeSC Majorana platforms[25]. The native impurities investigated in this work promise Majorana tunability in LiFeAs, inspiring other controllable strategies which can be explored soon. Since the atom manipulation by an STM tip is routinely performed[46], it could be used for artificial design of MZM patterns in a vortex lattice by planting impurities to selected vortices. Alternatively, we also propose a new tuning scheme that facilitates a fast, on-demand tunable Majorana device that could be fabricated by combining electrostatic gating effect[47] and piezoelectric stress[48] with LiFeAs superconductor (Fig. 4f). In particular, an in situ piezo-stressed STM measurement on LiFeAs[49] reported strong evidence for a smectic order with $C_4$ symmetry breaking, indicating the feasibility of the device proposed here for tuning MZM under proper chemical potentials. The creation and annihilation of MZM could be then electrically controlled with a fast manipulation rate, realizing one of the necessary conditions for non-Abelian Majorana braiding and fault-tolerant topological quantum computation (see more details in the Methods section).

## Methods

**Single-crystal growth**. High-quality single crystals of LiFeAs were grown using the self-flux method[26]. The precursor of $Li_3As$ was first synthesized by sintering Li foil and an As lump at ~650 °C for 10 h in a Ti tube filled with argon (Ar) atmosphere. Then the $Li_3As$, Fe and As powders were mixed according to the elemental ratio of $LiFe_{0.3}As$. The mixture was put into an alumina oxide tube and subsequently sealed in a Nb tube and placed in an evacuated quartz tube. The sample was heated to 1100 °C for 20 h and then slowly cooled down to 750 °C at a rate of 2 °C per hour. Crystals with a size of up to 5 mm were obtained. To protect the samples from reacting with air or water, all the synthesis processes were carried out in a high-purity Ar atmosphere.

**Scanning tunneling microscopy measurements**. STM/S measurements were conducted in an ultrahigh vacuum ($1 \times 10^{-11}$ mbar) USM-1300-$^3$He system with a vector magnet. The energy resolution is better than 0.28 meV. Tungsten tips were calibrated on a clean Au(111) surface before use. To protect the samples from reacting with air or water, LiFeAs sample for STM measurements was mounted in a glove box filled with high-purity Ar atmosphere (>99.999%). After a quick transfer to STM chamber, it was cleaved in situ at room temperature and transferred to the scanner immediately. Vertical magnetic fields were applied to the sample surface. All data shown in this paper were acquired at 400 mK. STM images were obtained in the constant-current mode. Differential conductance ($dI/dV$) spectra and constant bias maps were acquired by a standard lock-in amplitude at a frequency of 973.0 Hz under a modulation voltage $V_{mod} = 0.1$ mV. All the data acquired by STM/S in this work (except for the barrier dependent measurements shown in Supplementary Fig. 1c) was measured under the same setpoints: sample bias $V_b = -5$ mV; tunneling current $I_t = 200$ pA. As the tip-sample separation is sufficiently large, the observed atomic-resolved features in topography correspond to lithium atoms. If the tunnel current was considerably increased in experiments, the atomic-resolved topography could show the sites of arsenic atoms[38]. The zero-bias conductance peaks were reproduced in nine impurity-assisted vortices (impVs) that are measured in four independent samples and with four different tips.

**Issue of inhomogeneity in FeSC Majorana platforms**. LiFeAs is the most homogeneous material among all the existing topological iron-based superconductors in which vortex MZMs are realized. As for Fe(Te,Se)[2] and (Li,Fe) OHFeSe[16], the topological band structure requires elemental substitutions which induces bulk inhomogeneity; and for another compound $CaKFe_4As_4$[22], although it has a stoichiometric bulk, which realizes the fully-theoretical-reproduced spatial patterns of integer-quantized vortex bound states, the cleavage surface is polar and suffers from surface inhomogeneity. Those features complicate the application of MZMs. LiFeAs has a layered structure (Fig. 1a), and it can be cleaved in between the lithium layers, resulting with a charge neutral surface of lithium atoms. Thus, LiFeAs offers a great promise for creating, manipulating, and tuning MZMs, owing to the homogeneous conditions observed both in the bulk and on the surface.

**Origins of vortex bound states in freeVs**. To reveal the origin of the two classes of differently behaving vortex bound states in freeVs, we measure the vortex shape by zero-bias conductance map. We find that the low energy quasiparticles have a star-like shape with the tails along the Γ-X direction (Fig. 2a). This anisotropy is likely caused by the rounded-square Fermi surface of outmost $d_{xy}$ orbital, where the parallel flat segments of Fermi surface are perpendicular to Γ-X direction[27,50–56]. Furthermore, it has been resolved clearly in LiFeAs that the larger (smaller) superconducting gap $\Delta_1$ ($\Delta_2$) opens on the inner $d_{yz}$ (outer $d_{xy}$) Fermi pocket, which has smaller (larger) $E_F$[55]. Hence the $d_{yz}$ ($d_{xy}$) orbital related vortex bound states have a larger (smaller) level spacing due to which it is easier (harder) to approach the quantum limit[39], thus appearing as discrete (dispersive) bound states.

**Reproducibility of MZMs observation in impVs**. We have repeated the observation of spatially non-splitting ZBCPs in nine different impVs (Fig. 3 and Supplementary Fig. 4) and checked all the necessary aspects carefully. We first determined that the sub-gap states shown in Fig. 3c are not impurity bound states. We measured $dI/dV$ spectra under zero field at the same measurement positions as that for the vortex bound states under 2.0 T. This demonstrates clearly that the impurities do not introduce the zero energy bound states (Fig. 3d). A zero-bias conductance map further shows that the vortex area does not have the zero energy quasiparticles when under zero field (Supplementary Fig. 1d). Furthermore, we checked the magnetic field evolution of the impurity bound states at the impV location by carefully avoiding vortex pinning. We found the impurity bound states never turn out to be zero energy (not shown). In addition, the observed ZBCPs in an impV are robust against changing tip-sample distance. The ZBCP are stable at the zero energy over two orders of magnitude of tunneling barrier conductance (Supplementary Fig. 1c), fully consistent with the appearance of a single MZM.

**Evidence of electron doping effect around impurities**. Unlike the well-defined Fermi level in the clean regions[31], the areas with impVs may have different chemical potential ($\mu$). By measuring the shift of the $d_{xy}$ band top, we reveal that the impurities shift $\mu$ up towards the TDS crossing. It is known that the band top of the outmost $d_{xy}$ orbital appears as a hump in $dI/dV$ spectrum at ~+33.4 meV (the black curve in Fig. 5a and Supplementary Fig. 2). This band top position provides an indicator for $\mu$ variations at different positions. While the band top hump position is fixed across the clean region (Supplementary Fig. 2), it shifts to lower energies in the vicinity of an impurity (red curve in Fig. 5a), which indicates electron doping induced by the impurity in its surroundings.

We perform detailed measurements of the spatial variation of the band top hump across an impV. A ZBC map for the vortex, topography, and line-cut

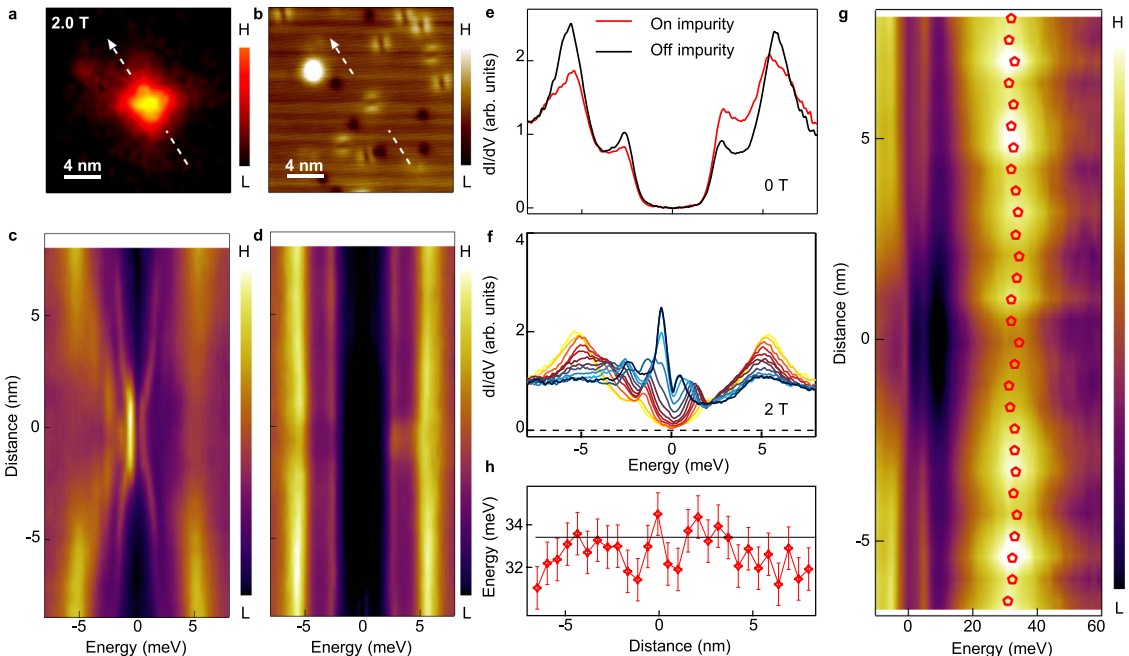

**Fig. 7 Ordinary vortex bound states in a weak impurity-assisted vortex. a** ZBC map around an impV. **b** Corresponding STM topography of **a**. The vortex is pinned to a weak impurity that is formed by $D_{2X}$ symmetric defects located on lattice sites. **c** Line-cut intensity plot measured under 2.0 T along the white line indicated in **a**, which demonstrates the spatial evolution of vortex bound states. **d** Line-cut intensity plot measured at 0 T and the same location as that in **c**, which demonstrates the spatial evolution of impurity bound states. **e** Zero field d$I$/d$V$ spectra measured at and away from the impurity site. **f** Selected spectra from **c**. **g** Wide range line-cut intensity plot of the impV with a linear background subtraction. The red symbols are the extracted energies of the band top following the method introduced in Fig. 5. **h** Spatial evolution of the band top energy across the impV. The error bars are defined as same as that in Fig. 5d.

measurement across the vortex are shown in Fig. 5e. To enhance the visualization of the hump and the variations of $\mu$, we perform linear background subtraction on each d$I$/d$V$ spectrum (Fig. 5b). Its corresponding raw data is shown in Fig. 5f. We note that the superconducting gap and even the MZM can be identified in these wide-energy-range measurements. To extract the relative $\mu$ shift reliably, we calculated the negative of the second derivative of the raw data shown in Fig. 5f. Subsequently, the hump energy at each spatial position was defined as the peak energy of negative second derivative spectra, which are extracted by a simple Gaussian fit (Fig. 5g). As shown in Fig. 5b, those extracted hump energies matched with the linear background subtracted intensity very well. We also measure the same line-cut under zero field (Fig. 5c), with the extracted hump energy showing good agreement (Fig. 5d) with 2.0 T results. We have performed the same procedure for 9 impVs in this study. Despite the different extent of tunability, we found that all of the 9 impurities shift $\mu$ upward (Fig. 6a and Supplementary Tables 1 and 2). The available $\mu$ range in impV cases is marked as the green shaded region in Fig. 4a, b.

**Evidence of impurity potential dependent tunability**. The scattering strength of impurities can be estimated from the energy of the impurity bound states, i.e. a stronger impurity can induce a lower energy impurity bound state[57]. We now turn to study the correlation between the energy of the lowest impurity bound states ($E_{IBS}$) and the multiple tuning effects of impurities. By measuring d$I$/d$V$ line-cut with and without magnetic field at the same positions (Figs. 3 and 7 and Supplementary Figs. 3 and 4), we successfully determined several featured quantities of the impVs simultaneously, such as the $\mu$-shift which is defined as the maximum hump energy deviation from the reference value in a clean region; the energy of the lowest non-zero vortex bound states ($E_{VBS}$) of impVs with ZBCPs ($E_{VBS}$ is also the topological gap which protects MZM); and the $E_{IBS}$ defined above. To reveal the underlying relationships, we measured several impVs following the strategies mentioned above and summarized them in Fig. 6a, b. First, we find that $\mu$-shift is anticorrelated with $E_{IBS}$ in the impVs (Fig. 6a). It leads to a reasonable conclusion that a stronger impurity induces a greater electron doping in its vicinity. Second, we also found an anticorrelation between $E_{VBS}$ and $E_{IBS}$ (Fig. 6b). It implies that stronger impurities are better for stabilizing MZM in an impV.

The behavior in Fig. 6 points to a possible scenario that stronger impurity introduces larger electron doping, pushing $\mu$ to be closer to the Dirac point of TDS phase (Fig. 4a, b). Subsequently, the value of $\Delta^2/E_F$ (for bulk Dirac cone) becomes larger in the vicinity of impurities, which satisfies a prerequisite for larger topological gap of MZM. This impurity-potential-dependent behavior demonstrates the tunability of vortex Majorana modes in LiFeAs.

**The case of non-ZBCP for weak impurities**. In our STM/S experiments, the ZBCP signature of MZMs does not appear in every impVs. We investigated the statistical distribution of vortices belonging to different classes by measuring all of the vortices in a selected area (150 nm by 150 nm). In this way we have determined that among the impVs, the probability of observing MZM is 14%. The absence of MZM in other impVs is likely related to the weakness of impurities to which the vortices are pinned as discussed below.

As summarized in Fig. 6, the degree of tunability due to an impurity is related to its potential strength. A stronger impurity induces a larger chemical potential shift, which leads to a more favorable condition for the emergence of vortex MZMs. Therefore, we note that the lack of MZM in some impVs is because the given impurity is too weak to sufficiently influence its vicinity. There are several aspects to the effect of the impurity strength. First, for weak impurities there is not enough electron doping in the surrounding region, so the Fermi level is still crossing the bent, Rashba-like part of TI surface state regime, forming two Fermi pockets. In this case, the vortex bound states in the impV behave similarly to a freeV. Second, a medium strength impurity can elevate $\mu$ to the TDS phase, however there is not enough asymmetric stress induced in its vicinity, which cannot break the $C_4$ symmetry of the square lattice. In this case, two pairs of helical Majorana modes emerge from the fully symmetric TDS phase. The featureless LDOS of helical Majorana modes, emerging from the $p_z/d_{yz}$ bulk band that forms the bulk Dirac fermions, overlaps with a pair of non-zero energy dispersive vortex bound states, which are related to the underlying $d_{xy}$ bulk bands[31]. Thus, the spectrum still appears to be similar to that in a freeV, but with featureless helical Majorana modes hidden behind it. Finally, in the case of weak impurities the sphere of their influence is limited, so the shift of the chemical potential is restricted to a small region very close to the impurity. This does not allow for a formation of a sufficiently long topological vortex line and the second Majorana mode possibly emerges too close to the surface, fusing with the Majorana located there and in consequence removing the zero-bias conductance peak from the spectrum.

**Example of weak impVs without MZM**. As shown in Fig. 7, we perform detailed STM/S measurements on a weak impV without MZM. The line-cut measured under the zero field shows that impurity bound states appear just at the edge of the superconducting gap (Fig. 7d, e). It indicates the impurity is quite weak. Across the vortex center, we measure a line-cut intensity plot of the vortex. As shown in Fig. 7c, the spatial evolution of vortex bound states is very similar to that in a freeV shown in Fig. 2. Selected d$I$/d$V$ spectra are shown in Fig. 7f. We find the coupling of the weak impurity in impV does not change the main features of vortex bound states, although the intensity of vortex bound states is reduced in the impV as compared to the freeV shown in Fig. 2f, which is plotted with the same scale.

Furthermore, we performed the $\mu$-shift measurement on this impV (Fig. 7g, h) by the same methods introduced in Fig. 5. We observe that the $\mu$-shift here is smaller ($2 \pm 1$ meV). Those observations are fully consistent with the scenario that the insufficient strength of weak impurities cannot result in the appearance of MZM in weak impVs.

**2D lattice model and theoretical simulation.** To perform the calculations corroborating the analysis of the experimental data we use a tight-binding model of a 2D Dirac fermion described by the following Bogoliubov-de Gennes Hamiltonian:

$$H = \sum_j c_j^\dagger \left( 2v\tau_z s_z - \mu\tau_z + \Delta(j_x, j_y)\tau_x + V_0 \exp\left(-\frac{((j_x - j_{x0})^2 + (j_y - j_{y0})^2)}{2\sigma^2}\right)\tau_z \right) c_j$$
$$+ \left( c_{j+\hat{x}}^\dagger \left(-i\frac{v}{2}\tau_z s_y - \frac{v}{2}\tau_z s_z\right) c_j + c_{j+\hat{y}}^\dagger \left(i\frac{v}{2}\tau_z s_x - \frac{v}{2}\tau_z s_z\right) c_j + H.c. \right) \tag{1}$$

where $j = (j_x, j_y)$ is the index labeling sites on the 2D square lattice, $c_j$ are the annihilation operators acting in Nambu space, $\tau_i$, $s_i$ are the Pauli matrices acting in particle-hole and spin spaces, respectively, $v$ is the velocity of the Dirac fermion, $\mu$ is the chemical potential,

$$\Delta(j_x, j_y) = \Delta_0 \tanh\frac{\sqrt{j_x^2 + j_y^2}}{\xi_0} \exp\left(i \arctan\frac{j_y}{j_x}\right) \tag{2}$$

is the superconducting order parameter with a vortex at $j = (0, 0)$, $\xi_0$ is the superconducting coherence length, $V_0$ is the strength of an impurity placed at $j_0 = (j_{x0}, j_{y0})$ described by a Gaussian potential. In such a model we calculate the eigenvalues $E_n$ with corresponding wavefunctions that are composed of the particle components $u_{n,\sigma}$ and hole components $v_{n,\sigma}$. With these eigenvalues and wavefunctions we compute the local density of states given by:

$$\rho(E) = -\sum_{n,\sigma} |u_{n,\sigma}|^2 f'(E_n - E) + |v_{n,\sigma}|^2 f'(E_n + E) \tag{3}$$

where $f'(E)$ is the derivative of the Fermi-Dirac distribution. The parameters used in the calculation are: $v = 1, \mu = 0.0475, \Delta_0 = 0.025, V_0 = -0.21, j_{x0} = j_{y0} = -22, \sigma = 6$. The width of the Gaussian scattering potential was chosen to correspond to the spatial extent of the impurity cluster as determined by the topography measurement. Based on the fit to the analytical wavefunction[11,43] (Fig. 3g), same as the fit performed in refs. [2] and [22], we can translate these values to $\Delta_0 = 2.1$ meV, $\mu = -4$ meV, $\xi = 3.6$ nm. The negative chemical potential ($\mu = -4$ meV) indicates that the underlying topological bands of MZM have hole-like dispersion, which is consistent with our scenario (Fig. 4a, b and Fig. 3f that the side peaks are stronger at the occupied side). By the eigenvalues and wavefunctions of vortex bound states obtained above, we plot the local density of states along a line-cut through the impurity position (Fig. 8). To incorporate the extra broadening of vortex bound states observed in Supplementary Fig. 1a, b, we model the broadening using effective temperature of the results with FWHM of 0.5 meV for zero energy mode and 1 meV for the higher energy states, comparable to the measured value (Fig. 3h–j).

**Characteristics of strong and weak impurities.** In the previous sections, we demonstrated that a vortex MZM can only be induced in an impV if it is coupled to a strong impurity. Here we show more detailed characterization of the configuration (topography) and bound states (d$I$/d$V$ spectra) of the impurities and vortices coupled to them, both for the cases with and without MZM.

In Supplementary Figs. 3 and 4, we show data for additional impVs studied in this work. For each impV, we measured the topography, ZBC vortex mapping, line-cut of vortex bound states under magnetic field and line-cut of impurity bound states under zero field. The spectra measurements with and without magnetic field were performed at the same positions along the line indicated in each topography. This enables a direct comparison of the influence of impurities on vortex quasiparticle excitations in different vortices. After performing data analysis as discussed above and in the main text, we summarize the extracted parameters for all of the Majorana impVs and ordinary impVs in Supplementary Table 1 and Supplementary Table 2, respectively.

On the surface of as-cleaved LiFeAs, we identify two classes of native impurities. The most distinguishable one is simple impurities for which their symmetry, configuration and lattice position can be well resolved by STM. Such simple impurities have been studied intensively in the literature[35–37]. As shown in Supplementary Figs. 3e, we measured six different kinds of simple impurities with their symmetry marked at the top of each panel. Our identification of those impurities is consistent with previous works[35–37]. However, even though those simple impurities have different symmetry, configuration and occupied positions, our experiments show that the impVs coupled to a single simple impurity cannot induce the vortex MZM (Fig. 7 and Supplementary Fig. 3a, d), suggesting that the single simple impurities are too weak to sufficiently change the material properties in their vicinity.

The other class of impurities has a more complex appearance in STM topography. These are either large clusters with large height and coverage (Fig. 3b, Supplementary Fig. 4a, d, and 4e–g) or groups of simple impurities crowding and combining in a small area, enhancing their influence far beyond what could be achieved by a single impurity (Supplementary Fig. 4b, c, h). In this work, all of the Majorana impVs are due to complex impurities. Such complex impurities have a strong influence on their surroundings, and thus were named strong impurities in the above sections (Fig. 3 and Supplementary Fig. 4). It is difficult to identify the symmetry and exact occupied lattice positions of those complex impurities. Moreover, in impVs in which the MZM appear, the coupled strong impurities have various configurations. Those observations indicate that the emergence of vortex MZMs in these impVs is not directly related to the symmetry, position occupied, or specific configuration of the impurities.

In addition, we measured the $\mu$-shift of 9 impVs (both with and without MZMs) following the methods introduced in Fig. 5. The results are shown in Supplementary Tables 1 and 2 and Fig. 6a. Even though the impurities belong to various classes, they all introduce electronic doping of about meV. Such a small $\mu$-shift seems to be of a reasonable magnitude for the local impurities. Moreover, it is adequate for tuning the band structure of LiFeAs into different topological phase, as the separation between the two Dirac points is only as small as 15 meV.

**Strategies for controlling vortex MZMs by impurities.** In order to provide guidelines for designing controllable Majorana patterns by impurity planting, here we point out directly the required conditions for impurities to introduce vortex MZM. As demonstrated above, the most important aspect of impurities for inducing vortex MZMs is their potential strength, rather than their exact location or configuration.

For creating MZM in an impV, the impurity assisting vortex quasiparticle excitations should satisfy three conditions: (1) providing sufficiently large electron doping; (2) providing sufficiently large lattice strain that breaks the symmetry; (3) influencing a sufficiently large volume of the sample that provides enough separation length for stabilizing MZMs.

Our experiments indicate that more complex impurities have a stronger influence, thus have a greater probability to induce vortex MZMs. Therefore, to engineer MZM in a specific vortex, one may need to gradually plant more and more impurities in the area of the vortex core by STM. In this process, vortex MZMs may appear when the tuning effects are strong enough to satisfy the three conditions mentioned above.

**Strategy for Majorana braiding without real space movement.** We note that it is not necessary to move the MZM in real space for the realization of non-Abelian braiding. The methodology for creation/annihilation of MZMs is described in the new proposals for braiding MZM in the Hilbert space. In our proposal, large scale impurity manipulation is required to create vortices with MZMs only for the initial setup. We do not expect to physically move vortices around to perform braiding or other manipulation, and thus we do not require further movement of the impurity clusters.

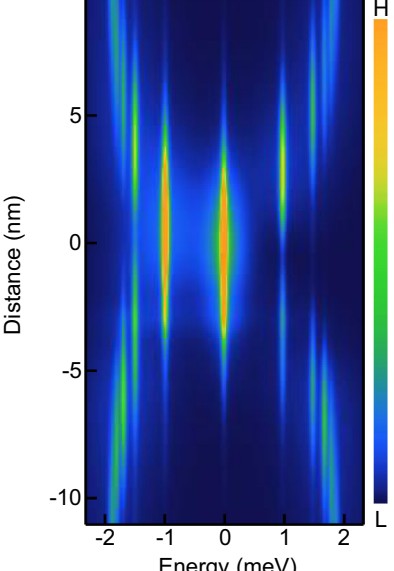

**Fig. 8 Model calculation of vortex bound states without applying extra broadening.** Without broadening, the non-zero energy bound states can be distinguished and no longer appear as a single dispersive peak in the spectrum.

We note that moving MZM in real space (with maintaining coherence) would be extremely difficult, not only in our system, but in any Majorana platform. This is one of the main reasons why the first proposal of braiding MZMs in real space[58] was soon updated by other strategies. The most up-to-date method for Majorana braiding is through the manipulation of the Hilbert space that is spanned by MZMs. The braiding process can be realized by sequentially controlling the tunneling coupling among MZMs, while those Majorana modes are still fixed in the real space[24,59–62]. These ideas can be employed in a Majorana lattice designed by impurity manipulation in LiFeAs and have a good potential for implementing a braiding method.

However, a specific design is beyond the scope of our manuscript. We are calling for more detailed and specific theoretical efforts that could be based on our experimental observations in this work.

## Data availability
The data that support the findings of this study are available from the corresponding authors on reasonable request.

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

## Acknowledgements

We thank X.-X. Wu, P. Zhang, and J.-Q. Lin for helpful discussions. This work at IOP is supported by grants from the National Natural Science Foundation of China (11888101, 61888102, 51991340, 11820101003, and 11921004), the Chinese Academy of Sciences (XDB28000000, XDB07000000) and the Ministry of Science and Technology of China (2016YFA0202300, 2019YFA0308500, 2018YFA0305800, and 2018YFA0305700) and Beijing Municipal Science & Technology Commission (No. Z191100007219012). The work at MIT is supported by DOE Office of Basic Energy Sciences, Division of Materials Sciences and Engineering under award no. DE-SC0019275.

## Author contributions

H.D. and H.-J.G. designed the experiments. S.Z. and L.C. performed the STM experiments with assistance from L.K., G.L., P.F., W.L., F.Y. and S.D.; M.P. and L.F. provided theoretical models and simulations. G.D., X.W. and C.J. provided samples. L.K. and L.C. analyzed experimental data with input from all other authors. L.K. plotted figures with input from all other authors. L.K., H.D. and M.P. wrote the manuscript with input from all other authors. All the authors participated in analyzing experimental data, plotting figures, and writing the manuscript. H.D. and H.-J.G. supervised the project.

## Competing interests

The authors declare no competing interests.
