## [Peer Review File · Nature Communications]

REVIEWER COMMENTS

Reviewer #1 (Remarks to the Author):

This paper by Lingyuan Kong et al. reports on the observation and interpretation of zero-bias peaks in the density of states of impurity pinned vortices in LiFeAs. By considering the topological nature of the underlying band structure the authors argue for these results to be signatures of Majorana quasiparticle excitations drawing parallels to similar recent reports in other iron based superconductors. The key advance to those works is that the material studied here is exceptionally clean and has a charge neutral surface that would in principle lend itself more easily to local manipulation of vortices. While the authors do not demonstrate such a manipulation they nevertheless demonstrate that LiFeAs shows properties similar to those reported in the other iron based superconductors and thereby provides the basis for such further studies. While the results therefore are probably not of interest to a general audience yet I believe that in principle they would 'represent important advances of significance to specialists within [the] field' as they open up the study of these phenomena in an effectively clean system and therefore qualify for publication in Nature Communications within the context of the previous publications by the group. In addition, the experimental work appears to have been carried out very carefully and systematically at a level one would expect from a publication in nature Communications.

While therefore recommending publication on grounds of interest and quality there are nevertheless several conceptual points that I would like to raise which I believe preclude publication unless satisfactorily addressed by the authors.

(1) Central to the paper are the topological aspects of the band structure of LiFeAs at the chemical potential where the superconducting gap opens. In figure 1b the authors are showing the projection of that bulk band structure on the surface of a (001) cleave referring to ref. 28. The band structure with the shown chemical potential is indeed the same as in Fig. 2d of ref 28. However the Fermi energy indicated in the present paper by the zero line in Fig. 1b seems to be the Fermi energy for approximately 3% Co doped samples in ref 28 with the Fermi energy for undoped samples being much lower at the top of the alpha band in a regime of the band structure with fundamentally different topological properties. Could the authors please comment?

Could they also clarify where the chemical potential would have to be in figure 1b for the cases discussed in figure 4?

In addition figure 1b should have an energy scale shown.

(2) The authors argue that in the vicinity of the impurities the chemical potential is being shifted by up to 4meV. Given the focus on tunability in the paper and that the energy distance between the TI and TDS points in the band structure are of the order of 20meV (ref 28) could the authors clarify what the key impact of such a small change is by identifying the shift in Fig. 1b and how it potentially could be enhanced if necessary?

(3) For the BdG equations it is relevant if the impurity site is a magnetic, a nonmagnetic potential or indeed mixed scatterer. Could the authors comment on the choice of scatterer and how the theoretical model changes when different types of pinning centres are being considered? Given the central importance for comparison with the experimental data could the authors further comment on the experimental evidence as to if their pinning centres are magnetic or charge neutral or indeed how such differences would impact the theoretical model and therefore conclusions?

(4) It is proposed in the conclusions that impurities can be manipulated. This is typically achieved in STM by manipulating adatoms on the surface. At a minimum the impurities should be in the top Li layer. Many impurities the impurities studied here have been attributed to locations in the underlying FeAs layer and therefore are not manipulatable by STM as with current techniques. As a minimum the authors should discuss where their pinning impurities are most likely located.

It seems unclear if adatoms on the Li layer would sufficiently couple to the FeAs layer in order to achieve the necessary chemical potential shifts. Given that this is part of the central conclusions could the authors comment in more detail on the scheme they envisage?

(5) Finally the authors should elaborate of what tunability exactly they believe is achievable as no tunability on the relevant energy scales of the band structure seems to have been demonstrated here (see also point above about unmovable impurities in buried layers).

(6) A minor point is that almost all 2D color plots (e.g. topographs etc) are missing colorbar scales.

On balance I feel that unless the above points are addressed satisfactorily I cannot overall recommend publication in Nature Communications.

Reviewer #2 (Remarks to the Author):

The observation of Majorana zero modes in vortices are of outstanding topical interest. The authors report on the investigation of such Majoranas by scanning tunneling microscopy and spectroscopy in one of the Fe-based superconductors, namely LiFeAs. Because of its layered structure, this material cleaves very well and is, therefore, well suited for STM. Making use of naturally occurring impurities the authors identify the corresponding zero-bias conductance peaks. Based on the band structure of the material, which is inferred from ARPES and exhibits two Dirac points, they also provide a phenomenological explanation for the observed differences in the vortex states close to impurities, as compared to vortices away from impurities. While the reported results appear to be obtained with sufficient care, I miss important pieces of information and appropriate citation of relevant literature.

1. A crucial issue for the whole manuscript is the distinction between free vortices and impurity-assisted vortices. However, there is no good evidence provided just how this distinction is made and, more specifically, under which conditions the different types of vortices are formed. As examples: Do the free vortices move when the applied field changes? Are the impurity-assisted vortices always show up at the same impurities, i.e. are they still at the same locations if the sample is warmed up above T_c and cooled down again? For me, it is curious that the vortex in Fig. 2b shows up at the position presented and not near one of the impurities close by. Could there be an impurity just below the surface, not visible in topography? Or is this vortex held in place by the surrounding ones (the distance should be about 30 nm, hence just outside the field of view). Conversely, what is the evidence that the impurity in Fig. 3b truly influences the vortex nearby, in other words why is the vortex located right at this impurity and not one of the neighboring ones, e.g. the big one at the top of Fig. 2b? At line 76, the authors claim a pinning of the vortices by the impurities. Here again, I would like to see what happens if one goes above T_c and cools down again. At the very least, field sweeps should be presented. In line of comment 3) below, the defects need to be characterized. Moreover, in order to claim "controllable strategies" and an "artificial design of MZM patterns in a vortex lattice by moving impurities" (lines 188 -190) the authors need to specify under which conditions which type of vortex is formed. Otherwise, one might move an impurity around which in the end does not affect a vortex (as seen for many impurities in Fig. 2b).

2. It has been pointed out (Kreisel et al., PRB 94, 224518 (2016), not cited by the authors) that the conductance maps and topographs obtained by STM on LiFeAs is sensitive to experimental conditions such as setpoint bias, tip height and also the tip itself. This needs to be discussed in detail as it may have an impact on the differences observed here. Note that the authors of the above-mentioned paper explicitly refer to inhomogeneous superconductivity and STS spectra in this system.

3. In continuation of my two points above, it has been shown (Schlegel et al., Phys. Status Solidi B 254, 1600159 (2017), not cited by the authors) that some defects only show up under certain tunneling conditions. Hence, the authors of the present manuscript need to present more detailed data to rule out that there actually might be an impurity also, e.g. in their Fig. 2b (actually, I couldn't find the bias voltage for this image). In addition, and possibly even more important, the above-mentioned paper analyses in detail the different defects and their different impact on the superconducting state. Hence, the authors of the present manuscript need to evaluate the defects in much more detail. Can all of them create Majorana zero modes? Are the different symmetries of the defects of consequence for the results observed here? In the Extended Data section the authors talk about electron doping (starting at line 438); is this true for all types of defects? The authors themselves discuss differences in the impurities and mention that the Majorana zero modes do not appear in all impurity-assisted vortices (line 478); I don't understand why this is not related to the specific type of defect in question. Again, understanding the impact of the different defects is also of importance for the proposed tuning scheme.

4. In Fig. 3g and lines 104-7, the authors discuss an asymmetry of the zero-bias conductance across the impurity-assisted vortex. How does this depend on the orientation along which the cut is taken (i.e. the orientation of the dashed line in 3a)? The authors only relate this vaguely to the impurity (line 112-3). The anisotropy, however, could also be related to the underlying orientation of the crystallographic lattice or, importantly, to the symmetry of the specific defect? How does it depend on applied field?

5. The authors discuss the zero-bias conductance peaks and the higher levels of the vortex bound state, lines 117-122. This part is badly written and difficult to decipher. In addition, I wonder how the authors can rule out an impact of the impurity here. As an example, it has been shown for LiFeAs by low-T STS (Chi Ming Yim et al., Nat. Comm. 9, 2602 (2018), not cited by the authors) that local strain influences the local spectra massively. How can the authors of the present manuscript rule out some local strain close to the impurity which, in consequence, influences the tunneling spectra?

6. The samples need to be characterized better. At the very minimum, the two transition temperatures need to be given and compared to literature.

7. In case of free vortices, the authors refer to "a pair of dispersive vortex bound states" (lines 87-88). What is the justification for calling this a pair, or with other words why is the peak on the negative-bias side so much more pronounced compared to the positive bias, particularly within the vortex. Note that others (Chi Ming Yim et al., see comment 5. above) reported the feature at positive bias to possibly being even less developed as shown in the present manuscript (this is, however, difficult to infer from the plots provided).

8. In their concluding remarks, the authors propose a "tunable Majorana device that could be fabricated by combining electrostatic gating effect⁴⁰ and piezoelectric stress⁴¹ with LiFeAs superconductor" (lines ,also lines 361-2). Strained samples LiFeAs have already been investigated by low-T STM, see Chi Ming Yim et al., Nat. Comm. 9, 2602 (2018). A new CDW phase has been reported which may pose an issue for the purpose the authors of the present manuscript propose. Most

importantly, this highly relevant work is not cited. In addition, this work is also relevant for the discussed C4 symmetry breaking (lines 169-170) and the results of the present manuscript need to be discussed in light of this earlier study.

Minor remark:

Line 111: here the authors refer to Figs. 2c and 2d, yet this should be Figs. 3c and 3d, I guess.

Reviewer #3 (Remarks to the Author):

The manuscript from Lingyuan Kong et al reports observations/analyses of Majorana zero modes (MZMs) in a stoichiometric and homogeneous superconductor LiFeAs. This is clearly an extension of previous work on Fe(Se,Te)/(Li,Fe)OHFeSe from the same group. The major claims in this work are: 1. first observation of MZMs in impurity assist vortex (impV); 2. asymmetry of zero-bias conductance peaks(ZBCPs) across the impV indicates their topological origin; 3. a method to stabilize MZMs by tuning bulk Dirac fermion. My comments to these claims are as follows:

1. MZMs have been observed in Fe-doped LiFeAs (S.S. Zhang et al., Phys. Rev. B 101, 100507(R) (2020)). So the authors' first claim is not applicable.

2. The analysis of the asymmetric ZBCPs is not convincing yet. As shown in Fig. 3a, difference line cut of the conductance map may give rise to different distance evolution of the ZBCPs (Fig. 3g). Meanwhile, the comparison between Fig. 3g and Fig. 3j, i.e. experimental data and calculation can not support their claim. The degree of asymmetry and the deviation between experimental data and calculations are at the same level to me. A main signature in the theory (the shoulder in Fig. 3j at -3nm) is not clear in the experimental data.

3. Although the idea of stabilizing MZM by controlling chemical potential is not new (e.g. P. Zhang et al., Nature Physics 15, 41-47(2019)), the authors demonstrate that LiFeAs is an ideal platform to tune the MZM. This will generate broad interest and stimulate further research towards braiding MZM.

This work generates interesting topics and demonstrates high potential of LiFeAs in terms of manipulating MZMs. However, as addressed above, the novelty of this work is challenged by several facts. Therefore, I will leave the editor to decide whether it is suitable for Nat. Commun.

Summary of the Changes

The revisions are marked in blue (changes) or green (rearrangement) colors in the main text and supplementary information. The format of the manuscript was adjusted to fit the requirements of *Nature Communications*. Following the constructive suggestions of the reviewers, we revised our manuscripts as follows.

1. The abstract and introduction were adjusted to fit the formatting requirements of *Nature Communications*. We did not change the underlying meaning of the two parts.
2. We added headings and subheadings in the main text to fit the formatting requirements of *Nature Communications*.
3. We rewrote the section about dispersive high energy bound states in the main text following the suggestions of the Reviewers.
4. We added a discussion of *Nat. Commun.* **9**, 2602 (2018) in the last paragraph of the main text.
5. We added the energy scale in Fig. 1b
6. We added a discussion of *Phys. Rev. B* **94**, 224518 (2016) in the Methods.
7. We moved the Extended Data Figs. 1 and 3 of the previous version to a newly added Supplementary Information.
8. We added more detailed characteristics of impurity-assisted vortices and their impurities in Supplementary Figs. 3 and 4. The individual behaviors of impurity-assisted vortices are shown in newly-added Supplementary Tables 1 and 2. Two related new sections were added in the Methods, elaborating which kinds of impurities could create vortex MZMs and what conditions are required for creating vortex MZM.
9. We cited 8 more references in the revised manuscripts.
10. We corrected some typos and adjusted numbering accordingly.

We highly appreciate the constructive and stimulating questions from all the reviewers, which has clearly improved our work and strengthened this manuscript.

To Reviewer #1

We greatly appreciate the thorough consideration and high evaluation of our work by Reviewer #1. After carefully analyzing the referee's suggestions and comments, we have addressed all the issues raised by the reviewer, such as the concerns about the band structure, model calculation and more detailed features/effects of the impurities. To do so, we added more data and descriptions of the behavior of impurity-assisted vortices in the revised manuscript. We have also expanded the discussion of the model calculations. Below we answer all the questions in more detail point by point.

Reviewer #1 (Remarks to the Author):

This paper by Lingyuan Kong et al. reports on the observation and interpretation of zero-bias peaks in the density of states of impurity pinned vortices in LiFeAs. By considering the topological nature of the underlying band structure the authors argue for these results to be signatures of Majorana quasiparticle excitations drawing parallels to similar recent reports in other iron based superconductors. The key advance to those works is that the material studied here is exceptionally clean and has a charge neutral surface that would in principle lend itself more easily to local manipulation of vortices. While the authors do not demonstrate such a manipulation they nevertheless demonstrate that LiFeAs shows properties similar to those reported in the other iron based superconductors and thereby provides the basis for such further studies. While the results therefore are probably not of interest to a general audience yet I believe that in principle they would 'represent important advances of significance to specialists within [the] field' as they open up the study of these phenomena in an effectively clean system and therefore qualify for publication in Nature Communications within the context of the previous publications by the group. In addition, the experimental work appears to have been carried out very carefully and systematically at a level one would expect from a publication in Nature Communications.

While therefore recommending publication on grounds of interest and quality there are nevertheless several conceptual points that I would like to raise which I believe preclude publication unless satisfactorily addressed by the authors.

#Question:

1-1. Central to the paper are the topological aspects of the band structure of LiFeAs at the chemical potential where the superconducting gap opens. In figure 1b the authors are showing the projection of that bulk band structure on the surface of a (001) cleave referring to ref. 28. The band structure with the shown chemical potential is indeed the same as in Fig. 2d of ref 28. However, the Fermi energy indicated in the present paper by the zero line in Fig. 1b seems to be the Fermi energy for approximately 3% Co doped samples in ref 28 with the Fermi energy for undoped samples being much lower at the top of the alpha band in a regime of the band structure with fundamentally different topological properties. Could the authors please comment?

#Answer:

We thank the referee for the raising this important point that we want to now clarify. It is well

known that while *ab initio* calculations capture the qualitative features of the electronic structure of FeSCs, they often do not accurately describe many important near-Fermi-level properties (such as the position of the chemical potential itself) of FeSCs in general (C. R. Phys. 17, 140 (2016)) and LiFeAs in particular (PRB 85, 094505 (2012); PRL 109, 177001 (2012)). In order to correct for such deficiencies and present the experimentally relevant topological band structure which is important for our discussions in Fig. 4, we have adjusted the chemical potential of the *ab initio* calculation for LiFeAs according to the ARPES measurements reported in Fig. S3(a) of Nat. Phys. 15, 41 (2019). Moreover, a modified *ab initio* calculation for LiFeAs (from the same group of Nat. Phys. 15, 41 (2019)) was shown in PRL 123, 217004 (2019) and PRB 101, 100507 (2020), in which the chemical potential is similar to what was shown in Fig. 1b of our manuscript.

1-2. Could they also clarify where the chemical potential would have to be in figure 1b for the cases discussed in figure 4?

#Answer:

The chemical potential of the case of Fig. 4c (Fig. 4d and 4e) is below (above) the Fermi level in Fig. 1b. The band bending feature of TI surface states is not clear in the previous *ab initio* calculation, which requires more careful consideration in the future. The cases of Fig. 4d and 4e share a similar chemical potential but require different symmetry breaking conditions as discussed in Fig. 4.

1-3. In addition, figure 1b should have an energy scale shown.

#Answer:

We added the energy scale in the revised figure.

#Question:

2. The authors argue that in the vicinity of the impurities the chemical potential is being shifted by up to 4 meV. Given the focus on tunability in the paper and that the energy distance between the TI and TDS points in the band structure are of the order of 20 meV (ref 28), could the authors clarify what the key impact of such a small change is, by identifying the shift in Fig. 1b, and how it potentially could be enhanced if necessary?

#Answer:

As the band structure of LiFeAs contains multiple bands in the energy region around the Fermi energy, there may be additional hybridization occurring between these states. As seen in the calculated band structure, for the intrinsic value of chemical potential both discussed Dirac cones are overlapping with d-orbital bands present at the same momenta. However, when the chemical potential is shifted, the states from various bands can become more clearly separated in momentum space. Therefore, even a small change of chemical potential of about 5 meV can be substantial. Moreover, a change of 5 meV in the chemical potential due to doping by the impurity is about 50% of the distance between the original Fermi energy and the Dirac point. This will thus have a substantial effect on the spacing of the vortex bound states as their energy separation scales like Δ/μ . The enhancement may potentially be achieved in two ways. First of all, by finding or engineering impurities with a stronger potential. Second, it could be potentially realized by electrostatic gating and applying extrinsic pressure as mentioned in Fig. 4.

#Question:

3. For the BdG equations it is relevant if the impurity site is a magnetic, a nonmagnetic potential

or indeed mixed scatterer. Could the authors comment on the choice of scatterer and how the theoretical model changes when different types of pinning centres are being considered? Given the central importance for comparison with the experimental data could the authors further comment on the experimental evidence as to if their pinning centres are magnetic or charge neutral or indeed how such differences would impact the theoretical model and therefore conclusions?

#Answer:

We agree with the referee that the type of the impurity is very important in considering bound states in Bogoliubov-de Gennes formalism. One major effect that a magnetic impurity can have in contrast to a nonmagnetic one is introducing a gap into the Dirac fermion spectrum. If the magnetic impurity potential is sufficiently strong, the chemical potential may end up inside of the gap and then no MZM would be observed. However, for mixed impurity types there is an intermediate regime where the magnetic perturbation is not strong enough to destroy Majorana mode and thus the qualitative conclusion will remain the same as for purely non-magnetic impurity.

By measuring the impurity bound states under different magnetic fields and carefully avoiding vortex pinning, we found that most of the impurity bound states are stable under changing magnetic field, while only some impurity bound states slowly vary their energy. This indicates that the impurity potential is most likely a mixed one with the non-magnetic component dominating. As the purpose of our simple theoretical model is to qualitatively demonstrate that the incorporation of impurities in a Majorana vortex will lead to anisotropic wavefunction, we have decided to use a purely non-magnetic impurity in order to decrease the number of fitting parameters of the theory (the value of impurity potential strength is very difficult to determine experimentally). Previous works (e.g., PRB 94, 224518 (2016)) have also adopted this approximation and have reached a good agreement between the theory and the experiment. However, we have checked that inclusion of a small to medium magnetic component does not significantly change the outcome discussed in the manuscript. Nevertheless, using a more detailed model of the impurity, including addition of a magnetic component of the scattering potential, will improve the quantitative agreement with the experiment and may be considered in future works.

#Question:

4. It is proposed in the conclusions that impurities can be manipulated. This is typically achieved in STM by manipulating adatoms on the surface. At a minimum the impurities should be in the top Li layer. Many impurities the impurities studied here have been attributed to locations in the underlying FeAs layer and therefore are not manipulatable by STM as with current techniques. As a minimum the authors should discuss where their pinning impurities are most likely located. It seems unclear if adatoms on the Li layer would sufficiently couple to the FeAs layer in order to achieve the necessary chemical potential shifts. Given that this is part of the central conclusions, could the authors comment in more detail on the scheme they envisage?

#Answer:

The most important aspect of impurities for inducing vortex MZMs is their potential strength, not their exact location. Among all of the impurity-assisted vortices (*impVs*), the *impVs* pinned by single weak impurities did not support MZMs (shown in the newly added Supplementary Figs. 3-4 and the related contents in Method in our revised manuscripts). The configuration and symmetry of those simple impurities can be well identified by our experiments, as shown in the figures attached (Fig. R1.1). Similar identifications can also be found in the literature, see Phys. Status Solidi B 254, 1600159 (2017); Nat. Commun. 8, 15996 (2017), and etc.

In contrast, the *impVs* with MZM are pinned by stronger impurities which are not single impurities. For example, such impurities appear as large clusters with large covering and height in STM topography (Fig. 3b). Such impurities, even if they are located in the top lithium layer, will introduce strong electron doping and lattice strain not only in the closest neighborhood, but will also affect larger volume of the sample, including FeAs layer beneath the surface. Thus, a large cluster of impurities can be planted at a selected vortex by systematic atomic manipulation using STM. Our results in the current work indicate that such manipulation can potentially turn on/off the MZM in the vortex.

Fig. R1.1 Typical simple impurity on LiFeAs surface. The arrows indicate nearest Fe-Fe direction. Scanning area: 3 nm by 3 nm. The tags marked on the top indicate the symmetry of each impurity, and the letters at the subscripts of tags indicate the direction of symmetry axis, e.g. D_{4MX} stands for a D_4 symmetric impurity with its symmetry axis along Γ -X and Γ -M.

#Question:

5. Finally, the authors should elaborate of what tunability exactly they believe is achievable as no tunability on the relevant energy scales of the band structure seems to have been demonstrated here (see also point above about unmovable impurities in buried layers).

#Answer:

In the manuscript, we proposed two methods for tuning the properties of this material. The electrostatic gating STM measurements have been performed in $(\text{Li,Fe})\text{OHFeSe}$ (Ref. 47), and the same technique can be also applied in our material. We also discussed the feasibility of the method of atomic manipulation mentioned in the answer of question #4.

In addition, in the past few months after submitting the current work, we found that external mechanical pressure can modify the chemical potential of LiFeAs up to several tens of meV. We have finished ARPES measurements of the pressure-dependent band structure and will publish it as a separate work. The pressure device used in the ARPES measurements can be adopted for future STM experiments, and some experiments proposed in the current work for directly realizing tunability of vortex MZM on LiFeAs can be performed in the near future.

#Question:

6. A minor point is that almost all 2D color plots (e.g. topographs etc) are missing colorbar scales.

#Answer:

We thank the referee for noticing the missing colorbars. In those 2D color plots, the exact values of the definite colors provide little information as the data is normalized. Therefore we only provide the indication on high and low sides, following the routine used in many previous works.

On balance I feel that unless the above points are addressed satisfactorily I cannot overall recommend publication in Nature Communications.

To Reviewer #2

We sincerely appreciate the thorough review and constructive suggestions of Reviewer #2. Following the reviewer's suggestions, we provide more information on impurity-assisted vortex (*impV*) in this reply and revised manuscript, by adding some new experiments results. With that, we further clarify the emergence mechanism of MZM in *impVs*. In addition, we carefully discuss the vortex dynamics, pinning, their behavior under varying conditions of temperature and magnetic field, and add more relevant references. We provide a point-by-point answer in the following.

Reviewer #2 (Remarks to the Author):

The observation of Majorana zero modes in vortices are of outstanding topical interest. The authors report on the investigation of such Majoranas by scanning tunneling microscopy and spectroscopy in one of the Fe-based superconductors, namely LiFeAs. Because of its layered structure, this material cleaves very well and is, therefore, well suited for STM. Making use of naturally occurring impurities the authors identify the corresponding zero-bias conductance peaks. Based on the band structure of the material, which is inferred from ARPES and exhibits two Dirac points, they also provide a phenomenological explanation for the observed differences in the vortex states close to impurities, as compared to vortices away from impurities. While the reported results appear to be obtained with sufficient care, I miss important pieces of information and appropriate citation of relevant literature.

#Question:

1. A crucial issue for the whole manuscript is the distinction between free vortices and impurity-assisted vortices. However, there is no good evidence provided just how this distinction is made and, more specifically, under which conditions the different types of vortices are formed.

#Answer:

We thank the referee for raising many important questions about vortex dynamics that we would like to answer in more detail. Since vortices are dynamic quantum objects, their pinning location is determined by the balance between the pinning force, the Coulomb repulsion and multiple other factors. The stable configuration of a group of vortices depends on the details of how the sample was cooled down and how the magnetic field is applied. Perturbations in temperature or magnetic field can change the vortex pinning locations.

Experimentally, we performed the measurements on the vortex configuration that have reached a stable state. The distinction between different types of vortices is then made by combination of spectroscopic and topographic measurements. First of all, we can see directly where the vortices appear and what kind of impurities they are pinned to. By then measuring the dI/dV spectra on each type of vortices, we can investigate the mechanism of vortex MZM appearance. Below we discuss all the detailed questions point by point.

As examples:

1-1. Do the free vortices move when the applied field changes?

#Answer:

Yes, the vortices can move if the applied field changes.

1-2. Are the impurity-assisted vortices always show up at the same impurities, i.e. are they still at the same locations if the sample is warmed up above T_c and cooled down again?

#Answer:

In general in the process of warming up (while remaining in superconducting phase), the thermal energy leads to thermal creep or drift of the pinned vortex. Thus, the vortices may move to different positions at a higher temperature. When cooling down again, thermal creep is usually forbidden, and the vortex will stay at the new location, different to the original one. However, in some special cases, it is also possible that the pinning force is strong enough to keep the vortex in place. If the temperature is ramping up slowly and smoothly, so that the thermal perturbation is small enough, the vortex will more likely stay at the same position.

When the sample is warmed up above T_c , vortices will disappear completely. When cooling down below T_c again, the positions at which the vortices are formed depend on multiple technical details, such as the specific process of cooling down and the local energy minimum of vortex. It is not guaranteed (but is possible) that the vortex will appear at the same position as before.

1-3. For me, it is curious that the vortex in Fig. 2b shows up at the position presented and not near one of the impurities close by. Could there be an impurity just below the surface, not visible in topography? Or is this vortex held in place by the surrounding ones (the distance should be about 30 nm, hence just outside the field of view).

#Answer:

Although vortices tend to be pinned by defects in a superconductor, the exact location of the vortex is a result of the balance between the pinning force of impurities and Coulomb repulsion of other vortices. Thus, a free vortex could appear occasionally. In our experiments, only a small fraction (about 17%) of the vortices are free vortices.

Furthermore, there are no invisible impurities appearing under the vortex shown in Fig. 2b. As shown below in the Fig. R2.3, we measured the zero-field dI/dV spectra along the same line shown in Fig. 2a&b. The zero-field spectra show a perfect and homogeneous behavior of the superconducting gap which indicates absence of impurity in the area.

1-4. Conversely, what is the evidence that the impurity in Fig. 3b truly influences the vortex nearby, in other words why is the vortex located right at this impurity and not one of the neighboring ones, e.g. the big one at the top of Fig. 2b?

#Answer:

The evidence for the influence of the impurity on the vortex is twofold. First, the local chemical potential shift can be measured near the vortex (an example is shown in Fig. 5 in the revised version). Second, the anisotropy of Majorana wavefunction is a direct result of the influence of the impurity. Experimentally, it can be recognized that the intensity of zero-bias conductance peaks (ZBCPs) is weaker at the location of impurity, visible as the local density of states encircling the impurity and signifying its strong impact on the vortex bound states. This is demonstrated clearly below in Fig. R2.2.

As we mentioned in the previous point, the location in which the vortex appears is a result of a balance. It depends on the specific process of cooling down and applying magnetic field. In a stable state under low temperature and magnetic field, we found the vortex located right at the impurity marked in Fig. 3b. We note that the vortex could also be located at the neighboring impurities, if sufficient perturbations, e.g. varying temperature or magnetic field, could cause the drift of the vortex to a new stable position.

1-5. At line 76, the authors claim a pinning of the vortices by the impurities. Here again, I would like to see what happens if one goes above T_c and cools down again. At the very least, field sweeps should be presented. In line of comment 3) below, the defects need to be characterized.

#Answer:

We thank the referee for this suggestion. We perform the field sweeps of vortex mapping in the same area (100 nm x 100 nm). As shown in Fig. R2.1, the vortices are rearranged under different magnetic fields, but some of them remain pinned to the same impurities when the field is increased.

Fig. R2.1 Vortex configuration under different magnetic fields. a, e, topography. b-d, zero-bias conductance mapping under 2 T, 4 T and 6 T respectively. The vortex positions belong to different magnetic fields are shown in e with different colors (white: 2 T; yellow: 4 T; green: 6 T).

In addition, we focus on a single vortex in a small area (Fig. R2.2a), and show here the zero-bias vortex mapping of three different field cycles (Fig. R2.2b-d). Each cycle consists of going between zero field and 2 T. As shown in the data, the pinning location of the vortex moves from the upper impurity to the lower one, and then back to the upper one, which is consistent with the mechanism of vortex movement we discussed above.

Fig. R2.2 Vortex location under different magnetic cycles. a, topography of the area. The circles with different color marked the vortex position under three different cycles (green: 1st cycle; white: 2nd cycle; blue: 3rd cycle). b-d, zero-bias conductance mapping of the vortex under the three magnetic cycles (2.0 T). The black square and cyan circle marked the positions of upper and lower impurity respectively.

1-6. Moreover, in order to claim “controllable strategies” and an “artificial design of MZM patterns in a vortex lattice by moving impurities” (lines 188 -190) the authors need to specify under which conditions which type of vortex is formed. Otherwise, one might move an impurity around which in the end does not affect a vortex (as seen for many impurities in Fig. 2b).

#Answer:

The most important aspect of impurities for inducing vortex MZMs is their strength, not their exact location or configuration of the impurities. For creating MZM in an *impV*, the impurity, assisting vortex quasiparticle excitations, should satisfy three conditions, i.e. the impurity (1)

provides sufficiently large electron doping; (2) provides sufficiently large lattice strain that breaks the C_4 symmetry; (3) influences a sufficiently large volume of the sample that provides enough separation length for stabilizing MZMs. Our experiments indicate that large clusters have such strong influence and can cause the appearance of the vortex MZMs. Therefore, for artificially creating MZM in a specific vortex, one may need to gradually plant more impurities in the area of the vortex core by STM. The vortex MZMs may then appear when the impurity potential is strong enough to satisfy the three conditions mentioned above. We added a new section of the Methods in the revised manuscript to elaborate on this issue.

#Question:

2-1. It has been pointed out (Kreisel et al., PRB 94, 224518 (2016), not cited by the authors) that the conductance maps and topographs obtained by STM on LiFeAs is sensitive to experimental conditions such as setpoint bias, tip height and also the tip itself. This needs to be discussed in detail as it may have an impact on the differences observed here.

#Answer:

We thank Reviewer #2 for pointing this relevant paper to us. We cited it in the revised manuscript and added some discussion in the Methods.

In that paper, the authors found that when measuring under small tip height, large sample bias and with a sharp STM tip, the maxima of topographic image intensity are located at the positions above the As atoms. Otherwise, the topography shows the positions of Li atoms as usual.

In our measurements, the tip height is large (Setpoints: $V_s = -5$ mV, $I_t = 200$ pA), which is even larger than the height in the simulation shown in Fig. 7g of the cited paper. As a result, we recognized the atomic-resolved topography as Lithium lattice in Fig. 1c. Moreover, we use the same setpoints among all the measurements shown in our manuscript. Thus the differences observed among cases are not artifacts caused by experimental parameters. On the other hand, our experiments show that the exact positions and configurations of the impurities is less relevant to the emergence of vortex MZM (see newly added Supplementary Figs. 3-4).

Exploration of the topography changes of different kinds of impurities is feasible by performing STM measurements under much larger I_t . However, such an investigation is beyond the scope of this work.

2-2. Note that the authors of the above-mentioned paper explicitly refer to inhomogeneous superconductivity and STS spectra in this system.

#Answer:

In the above-mentioned paper, the issues of inhomogeneous superconductivity and STS spectra were mentioned due to the discrepancy between simulations and experiments in the area of impurities. However, in the homogeneous area away from impurities, the simulations are well consistent with the experiments.

This point is fully compatible with the picture presented in our manuscript that the sparsely distributed impurities tune LiFeAs parameters locally with an otherwise homogeneous superconducting background. This helps us to demonstrate the underlying Majorana appearance mechanism in a vortex and possibly provide a pathway to tune vortex MZM in the homogeneous background by other methods such as gating and applying pressure.

Indeed, the homogeneous superconductivity in defect-free areas has been well studied in the literature (e.g. PRL 109, 087002 (2012)) and in the current work (Supplementary Fig. 2).

LiFeAs is the most homogeneous material among all the existing FeSC compounds hosting MZMs.

#Question:

3-1. In continuation of my two points above, it has been shown (Schlegel et al., Phys. Status Solidi B 254, 1600159 (2017), not cited by the authors) that some defects only show up under certain tunneling conditions. Hence, the authors of the present manuscript need to present more detailed data to rule out that there actually might be an impurity also, e.g. in their Fig. 2b (actually, I couldn't find the bias voltage for this image).

#Answer:

We cited this relevant paper in our revised manuscript. Indeed, some defects only show up under certain conditions. In order to make sure that there are actually no impurities in the area of a free vortex, we show here a dI/dV linecut measurement (Fig. R2.3) along the same line as indicated in Figs. 2a&2b. The zero-field spectra show perfect and homogeneous behavior of the superconducting gap which indicates absence of impurity in the area. In addition, the bias voltage for Fig. 2b is -5 mV.

Fig. R2.3 Zero-field spectra in the area of a free vortex. Line-cut plot (a) and the corresponding dI/dV spectra (b) under zero field. There are no signals of impurities can be resolved.

3-2. In addition, and possibly even more important, the above-mentioned paper analyses in detail the different defects and their different impact on the superconducting state. Hence, the authors of the present manuscript need to evaluate the defects in much more detail. Can all of them create Majorana zero modes? Are the different symmetries of the defects of consequence for the results observed here?

#Answer:

We added detailed characteristics of the impurity-assisted vortices (*impVs*) in the revised manuscript. The defects are carefully characterized by both topography and their impact on the superconducting state. More details are shown in the newly-added Supplementary Figs. 3-4 and the related section in Methods.

Our observations indicate that all the single simple impurities, e.g. the impurities identified in the above-mentioned paper, cannot create MZMs, while the more complex impurities give rise to vortex MZMs. Our observations also demonstrate that the appearance of vortex MZMs is not correlated to the specific symmetry of the defects. More detailed description is presented in the revised manuscript.

3-3. In the Extended Data section, the authors talk about electron doping (starting at line 438); is this true for all types of defects?

#Answer:

Yes. In our observations, all types of defects induce electron doping, but of different magnitude. More details are shown in the revised manuscript.

3-4. The authors themselves discuss differences in the impurities and mention that the Majorana zero modes do not appear in all impurity-assisted vortices (line 478); I don't understand why this is not related to the specific type of defect in question. Again, understanding the impact of the different defects is also of importance for the proposed tuning scheme.

#Answer:

We agree with the referee that the appearance of MZM depends on the properties of the impurity, in particular its strength. To expand upon this important issue, we added detailed characteristics of the *impVs* in the revised manuscript and related the behavior to the specific types of defects (see Supplementary Figs. 3-4 and Supplementary Tables 1-2).

In addition, in order to make the tuning scheme clearer, we added a section in the Methods entitled "Strategies for controlling vortex MZMs by impurities". We point out directly the required conditions for the impurities to induce vortex MZM in the newly added section.

#Question:

4-1. In Fig. 3g and lines 104-7, the authors discuss an asymmetry of the zero-bias conductance across the impurity-assisted vortex. How does this depend on the orientation along which the cut is taken (i.e. the orientation of the dashed line in 3a)?

#Answer:

The spatial distributions of the Majorana wavefunction are roughly shown in the zero-bias vortex mapping (Fig. 3a). The positions with brighter color scales correspond to stronger intensity in the dI/dV line profile (e.g. Fig. 3g). Thus, in principle, Fig. 3a contains the information of asymmetry of Majorana wavefunction along any selected orientation. Experimentally, a more reliable method for the line profile of the wavefunction is by extracting the zero-bias conductance from the dI/dV linecut measurement (Fig. 3e). As it is time consuming, we only measured the ZBC asymmetry along one selected orientation indicated in Figs. 3a&b. However, the ZBC asymmetry depends on other orientations can be qualitatively deduced from Fig. 3a. The ZBC asymmetry along the selected orientation shows no correlation with crystallographic lattice or any symmetries of the impurities.

4-2. The authors only relate this vaguely to the impurity (line 112-3). The anisotropy, however, could also be related to the underlying orientation of the crystallographic lattice or, importantly, to the symmetry of the specific defect?

#Answer:

As shown in the newly-added Supplementary Figs. 3-4, the anisotropy shows no correlation with the symmetry of the specific defect. The lattice symmetry regulates the vortex shape, creating long tails along the Γ -X direction as we observed in the case of *freeVs* (Fig. 2), thus the Majorana intensity anisotropy is not related to the lattice symmetry.

4-3. How does it depend on applied field?

#Answer:

As shown in the newly-added Supplementary Fig. 4, we measured *impVs* under three different magnetic fields. We do not observe any correlation with the field magnitude.

Generally speaking, for an isolated *impV*, if the magnetic field changes leave the vortex in the same spot, we expect no changes in the asymmetry of zero-bias conductance.

#Question:

5-1. The authors discuss the zero-bias conductance peaks and the higher levels of the vortex bound state, lines 117-122. This part is badly written and difficult to decipher.

#Answer:

We thank Reviewer #2 for pointing out this lack of clarity. We rewrote this part (please see the revised manuscript), which is now hopefully easier to follow.

5-2. In addition, I wonder how the authors can rule out an impact of the impurity here.

#Answer:

As we discussed in Fig. 4, the appearance of ZBCPs in *impVs* (Fig. 3) requires both electron doping and local strain introduced by the impurities.

Experimentally, we have confirmed that the bound states observed in line-cuts measured under a magnetic field are the vortex bound states and not just impurity bound states seen under zero field. For example, in Fig. 3, the dispersive vortex bound states (Fig. 3c) and impurity bound states (Fig. 3d) appear at different energies and spatial positions. This demonstrates that the observed non-zero energy states are vortex bound states (Fig. 3c) and not the impurity bound states. However, the presence of impurities most definitely affects the energy and spatial distribution of the higher energy states as we have seen from the simulation.

5-3. As an example, it has been shown for LiFeAs by Low-T STS (Chi Ming Yim et al., Nat. Comm. 9, 2602 (2018), not cited by the authors) that local strain influences the local spectra massively. How can the authors of the present manuscript rule out some local strain close to the impurity which, in consequence, influences the tunneling spectra?

#Answer:

We cited the Nat. Comm. paper in the revised manuscript. In that paper, the local spectra (most apparently, the size of the superconducting gap) were influenced strongly when the strain-induced smectic order appeared. In our experiments, no smectic order can be found in the vicinity of all the impurities. Thus it is reasonable to believe that the existing strain near the impurities in our work has much weaker influence on the spectra. Indeed, by checking the spectra measured around impurities and under zero field (Fig. 3d and Supplementary Figs. 3-4), we found that those impurities only suppress the superconducting coherence peak and induce bound states. At the same time, the magnitude of the superconducting gap is seldom influenced, which is in stark contrast with the observation shown in the Nat. Comm. paper mentioned above.

#Question:

6. The samples need to be characterized better. At the very minimum, the two transition temperatures need to be given and compared to literature.

#Answer:

We show the temperature dependent measurements of resistance (Left panel of Fig. R2.4) and magnetic susceptibility of our samples. The superconducting transition temperature (T_c) determined from the two measurements is 18 K, which is consistent with our previous results (Solid State Communications 148, 538–540 (2008)). In spectroscopic measurements we identify two superconducting gaps, $\Delta_1 = 2.7$ meV $\Delta_2 = 5.8$ meV, which are consistent with the literature results (Refs. 27 and 30)

Fig. R2.4 Transport measurements of LiFeAs. **a**, Temperature dependence of the resistance for LiFeAs. **b**, Temperature dependence of magnetic susceptibility for LiFeAs.

#Question:

7. In case of free vortices, the authors refer to “a pair of dispersive vortex bound states” (lines 87-88). What is the justification for calling this a pair, or with other words why is the peak on the negative-bias side so much more pronounced compared to the positive bias, particularly within the vortex. Note that others (Chi Ming Yim et al., see comment 5. above) reported the feature at positive bias to possibly being even less developed as shown in the present manuscript (this is, however, difficult to infer from the plots provided).

#Answer:

We thank the referee for bringing our attention to the imprecise statement in that section. The dispersive side peaks arise due to the overlap of many higher-energy vortex bound states that are merging due to large energy broadening. These higher energy states come in pairs of states at $\pm E$ that are the eigenstates of Bogoliubov-de Gennes Hamiltonian. However, the wavefunctions of the states at two opposite energies can be related to different Bessel functions and thus have different weight at the same position. This is one of the reasons why there is an asymmetry at positive and negative bias. The same type of asymmetry is also clearly visible in the simulation. For example, in the case of vortices in proximitized Dirac fermions, the energy side with more pronounced peaks at the vortex center depends on the position of the chemical potential with respect to the Dirac point. In the case of other types of bands, this will in turn be dependent of the sign of the band mass.

To be more explicit, the vortex bound states emerging from the hole-like bands have a stronger intensity at the occupied side (the negative bias in Unisoku STM configuration), while those from the electron-like bands have a stronger intensity at the unoccupied side (the positive bias here). In our work, the peaks observed in Fig. 2 are more pronounced on the negative-bias side. It is consistent with our conclusion that those bound states emerge from the d_{xy} hole-like bands.

In that Nat. Commun. paper, the dominant vortex bound states are shifted to the positive energy due to the applied pressure. This kind of vortex is coexisting with the smectic order, which is dramatically different from the free vortex shown in our manuscript.

#Question:

8. In their concluding remarks, the authors propose a “tunable Majorana device that could be fabricated by combining electrostatic gating effect [40] and piezoelectric stress [41] with LiFeAs superconductor” (lines, also lines 361-2). Strained samples LiFeAs have already been investigated by Low-T STM, see Chi Ming Yim et al., Nat. Comm. 9, 2602 (2018). A new CDW phase has been reported which may pose an issue for the purpose the authors of the

present manuscript propose. Most importantly, this highly relevant work is not cited. In addition, this work is also relevant for the discussed C4 symmetry breaking (lines 169-170) and the results of the present manuscript need to be discussed in light of this earlier study.

#Answer:

We thank Reviewer #2 for pointing us towards this relevant work. We cited this work in the revised manuscript and added some discussions of the relevant issues following the suggestions of the reviewer.

#Question:

9. Minor remark: Line 111: here the authors refer to Figs. 2c and 2d, yet this should be Figs. 3c and 3d, I guess.

#Answer:

We thank Reviewer #2 for the careful review. We corrected this typo in the revised manuscript.

To Reviewer #3

We sincerely appreciate the thorough review and constructive comments and suggestions of Reviewer #3. First we would like to stress that although a zero-energy impurity state was reported in LiFeAs recently, our work studied very different situation from that study. There are several key differences as shown below, which clearly demonstrates the novelty of the current work.

Reviewer #3 (Remarks to the Author):

The manuscript from Lingyuan Kong et al reports observations/analyses of Majorana zero modes (MZMs) in a stoichiometric and homogeneous superconductor LiFeAs. This is clearly an extension of previous work on Fe(Se,Te)/(Li,Fe)OHFeSe from the same group. The major claims in this work are: 1. first observation of MZMs in impurity assist vortex (impV); 2. asymmetry of zero-bias conductance peaks(ZBCPs) across the impV indicates their topological origin; 3. a method to stabilize MZMs by tuning bulk Dirac fermion. My comments to these claims are as follows:

#Question:

1. MZMs have been observed in Fe-doped LiFeAs (S.S. Zhang et al., Phys. Rev. B 101, 100507(R) (2020)). So the authors' first claim is not applicable.

#Answer:

We thank reviewer #3 for pointing out this article, which we now cite in the revised manuscript. That paper reported on the observation of zero-bias conductance peaks on Fe impurities. Our work has several key differences as compared to that paper, which establishes the novelty of our results.

First, our work reports on the first observation of vortex MZM on LiFeAs, in contrast to impurity bound states of the PRB paper. The absence of MZM in vortex of LiFeAs in multiple previous studies was widely regarded as a big puzzle in the FeSC Majorana platforms field, as even though the topological band structure was clearly observed in LiFeAs, no vortex MZM have been found. Moreover, the zero-bias conductance peaks observed on Fe impurities can not be simply regarded as the same thing as the MZMs in a vortex. The near zero energy impurity states can arise due to trivial mechanisms, or they may become MZMs only when the impurity induces the so-called 'quantum anomalous vortex', which is still under debate. The conclusion about the fundamental difference between the magnetic impurity based zero bias peaks and zero energy vortex bound states is further supported by a recent work (Phys. Rev. Lett. 126, 076802), which identifies the zero bias peaks at magnetic impurities of iron superconductor as Yu-Shiba-Rusinov states.

Second, our work provides a thorough understanding of the behavior of vortex bound states in LiFeAs. We explain why the MZMs are absent in the free vortices in LiFeAs and demonstrate how to induce the vortex MZM in this material. This led us to propose a new way to realize tunable MZMs, and open up a new pathway towards MZM braiding in FeSCs.

Finally, it is the first time that topological Dirac semimetal phase is incorporated in the Majorana quasiparticle excitations. We show experimental evidence that the emergence of vortex MZM is caused by tuning the bulk Dirac fermions.

#Question:

2. The analysis of the asymmetric ZBCPs is not convincing yet. As shown in Fig. 3a, difference line cut of the conductance map may give rise to different distance evolution of the ZBCPs (Fig. 3g). Meanwhile, the comparison between Fig. 3g and Fig. 3j, i.e. experimental data and calculation can not support their claim. The degree of asymmetry and the deviation between experimental data and calculations are at the same level to me. A main signature in the theory (the shoulder in Fig. 3j at -3nm) is not clear in the experimental data.

#Answer:

We thank the referee for bringing our attention to the lack of clarity in demonstrating the connection between Majorana asymmetry and the impurity. First of all, we would like to point out that the circles in Fig.3j are the experimental data and they clearly exhibit a shoulder at the position of impurity. To provide further evidence for that we include Fig. R3 (Supplementary Figure 1e), which demonstrates the impact of impurity on the wavefunction. This figure displays the line profile from Fig. 3g and 3j with more details (black curve). The center of the impurity (marked by the green bar) can be distinguished by the minimum of the intensity of the superconducting coherence peak (the blue curve). The Majorana wavefunction (the black curve) develops a shoulder at around -3 nm, same as the model simulation shown in Fig. 3j. Moreover, the position of the impurity coincides with the position of the non-zero energy subgap state at zero field (the red curve). All of this shows that the impurity directly influences the vortex core state and is responsible for the asymmetry of MZM wavefunction.

Fig. R3 Details of the line profile of a Majorana *impV*.

In our simulation, we use a simplified model to capture several main features of the vortex bound states that arise due to coupling with impurities, i.e., (1) spatial dispersion of the high energy vortex bound states (Fig. 3h), (2) asymmetry of the Majorana wavefunction and (3) a shoulder of the Majorana wavefunction in the vicinity of an impurity (Fig. 3j). We reproduced all of the main features of the vortex bound states observed in our experiments with some features matched very well quantitatively. In particular, the presence of a shoulder is not an intrinsic feature for all of the impurities but depends on their potential strength. The parameters for the particular results presented in Fig. 3j were chosen so that the shoulder matches the data, which required an increased strength of the impurity potential. This is also consistent with the experimental observation that only the strong impurities induce MZM in vortices. A better quantitative match between the theoretical simulations and experimental data requires more

realistic modeling of the impurity based on the measurements, which are difficult to perform accurately by STM at the current stage.

#Question:

3. Although the idea of stabilizing MZM by controlling chemical potential is not new (e.g. P. Zhang et al., Nature Physics 15, 41–47(2019)), the authors demonstrate that LiFeAs is an ideal platform to tune the MZM. This will generate broad interest and stimulate further research towards braiding MZM.

#Answer:

We thank Reviewer #3 for the positive evaluation of our work. Although some related ideas were proposed before, it is the first time that the experimental evidence is presented.

This work generates interesting topics and demonstrates high potential of LiFeAs in terms of manipulating MZMs. However, as addressed above, the novelty of this work is challenged by several facts. Therefore, I will leave the editor to decide whether it is suitable for Nat. Commun.

REVIEWER COMMENTS

Reviewer #1 (Remarks to the Author):

I would like to thank the authors for the very careful considerations of my remarks.

#Question 1-1: Thank you for the clarification. Would it be possible to note in the paper explicitly that the Figure is based on the data in the supplementary material of the reference?

#Question 1-2: OK

#Question 1-3: OK

#Question 1-4: OK

#Question 2 : OK. Would it be possible to indicate the shift in chemical potential in Fig. 1?

#Question 3 : I disagree with the statement that the weak shift in magnetic field necessarily indicates a weak magnetic component as the shift will depend on the relative local (exchange / SOC dominated) field vs the external field.

#Question 4 : The authors state that the relevant impurities are actually clusters / complex impurity arrangements (that on top of that are strongly coupled to the host compound). In addition at different parts of the manuscript they highlight the importance of the symmetry properties of such clusters for the stabilization of the proposed MZMs. Given that controlled impurity manipulation is a real space atom-by-atom process it seems extremely difficult to imagine that the necessary coherent movements of MZMs required for braiding can be carried out in this way.

#Question 5: It is undoubtedly true that strain and electrostatic doping can break C_4 symmetry / shift the average chemical potential. And indeed adding more and more atoms to a cluster of impurities might induce ZBCPs. However even if these are MZMs, it is not clear at all how homogeneous lattice deformations or atom-by-atom growth of an impurity can be utilized in order to create movable Majorana patterns for braiding. To be clear, I do not doubt that these are good strategies for the targeted creation / destruction of the observed ZBCPs but have concerns describing these as controlling MZM in the sense of enabling braiding as stated in the last sentence of the discussion.

Question 6: OK

New Questions raised by the reply:

Q1

In light of the new discussion including the characteristics of the ZBCP generating impurities the authors state that these are (exclusively?) clusters. This implies an extended rather than a point scattering potential as assumed in the BDG equations. In how far are the proposed theoretical models even applicable to the experimental situation? I.e. can the authors add a discussion of the relevant length scales?

Q2

In the new sections the authors imply and emphasize that strong scattering centres are necessary to observe MZMs as they are not observed in weak impurities. 'Strongness' seems to be characterized by lower E_{IBS} and greater electron doping (i.e. higher chemical potential shift). Given that 14% of impV showed ZBCPs and 9 ZBCPs are observed there should be of the order of 50 impV that do not show ZBCP/MZM 'which is likely to be related to their weakness'. Given that the paper is about the controlled stabilization of MZM in impV I would expect a more quantitative separation / statistics on what makes these 14% of impV special - i.e. would it be possible to show a distribution of E_{IBS} or the chemical shift across all impV and demonstrate that the observed MZM are indeed occurring in a special subclass of impV? Would it be also possible to clarify which data points in Fig. 6 correspond to ZBCP hosting impVs and which not in order to demonstrate that these really separate into a 'strong' and 'weak'?

regime?

Overall, I have two remaining key concerns:

(i) It remains difficult to see how the proposed methodology for creation/annihilation of ZBCPs is sufficient to claim that braiding is possible in this system.

(ii) The provided evidence that strength is the key driving parameter for the creation of ZBCP seems insufficient as it is not clear from the data presented that the remaining 86% of impV not showing a ZBCP are indeed significantly weaker. This is key to the manuscript as the capability of controlling the occurrence of ZBCP as a function of this strength is one of the major conclusions.

These two points relate directly to the key claims of the paper.

To be clear, the experimental data in itself reporting the observation of ZBCPs at complex impV sites seems impeccable and their properties consistent with reports on the existence of ZBCP that are possible realisations of MZMs in related compounds.

However, the authors themselves state in abstract and discussion that the key scientific advance of this paper is not the observation of these experimental signatures in themselves but rather that they occur in homogenous LiFeAs as a potential future platform for manipulating / braiding MZMs. Beyond the host system for these phenomena in this case being homogeneous, I feel not sufficient evidence is provided that with existing atom-by-atom manipulation a controlled creation of a ZBCP hosting cluster is reasonably feasible in the sense that (i) the necessary composition/structure/key characteristics of such clusters has not been sufficiently determined in order to distinguish them from non-ZBCP-hosting clusters and (ii) the controlled assembly is within the realm of existing atom manipulation techniques.

Reviewer #2 (Remarks to the Author):

The authors have, in principle, addressed all my former comments and questions in a reasonable fashion. I therefore recommend publishing the revised manuscript after the authors have responded to one (minor) remaining issue:

In my earlier report, I asked the question (1-3 in the authors' response) "Could there be an impurity just below the surface, not visible in topography". This question is not answered. In fact, I'm surprised by the vigor with which the authors rule out such buried defects (specifically when working with only one value of the bias voltage). The authors should be very specific to what depth they think they see which type of impurity in their topography.

Reviewer #3 (Remarks to the Author):

In the revised manuscript, the authors resolved most of my concerns. In terms of novelty, I agree that although some related ideas were proposed before, an experimental realization is significant. In addition, the proposed "tunable Majorana device" could generate new thinking in the field of topological quantum computation. Therefore, it is my pleasure to recommend this work to be published in Nature Communications.

~~~~~

## To Reviewer #1

~~~~~

We greatly appreciate the careful review and constructive comments of Review#1 in this round. The two remaining concerns are the feasibility of braiding in our system and the insufficient characterization of the ZBCP hosting cluster. Below we respond to these concerns by introducing the up-to-date braiding strategies and performing more detailed data analysis.

Reviewer #1 (Remarks to the Author):

I would like to thank the authors for the very careful considerations of my remarks.

#Question 1-1: Thank you for the clarification. Would it be possible to note in the paper explicitly that the Figure is based on the data in the supplementary material of the reference? **#Question 1-2:** OK; **#Question 1-3:** OK; **#Question 1-4:** OK

#Reply:

Following the reviewer's suggestion, we added a note in the corresponding figure caption.

#Question 2: OK. Would it be possible to indicate the shift in chemical potential in Fig. 1?

#Reply:

We made the revision as required in Fig. 1.

#Question 3: I disagree with the statement that the weak shift in magnetic field necessarily indicates a weak magnetic component as the shift will depend on the relative local (exchange / SOC dominated) field vs the external field.

#Reply:

We agree with the reviewer that the shift depends on the relative strength of local vs the external field, but the dominant non-magnetic scattering potential is a well-established phenomenon in many iron-based superconductors (including LiFeAs), demonstrated by other measurements such as magnetization.

It has been shown that even for deliberately introduced substitution impurities at iron sites, the scattering potential of Co, Cu impurities has no or a very weak magnetic component (*e.g.* Nat. Commun. 4, 2749 (2013)), only the Mn impurity is expected to be magnetic, but it is only weakly coupled to the itinerant electron spins and hence one does not expect their magnetic potential to strongly modify the LDOS (*e.g.* Sci. Rep. 4, 6252 EP (2014)). Previous studies of impurity bound states in LiFeAs have also shown a dominant nonmagnetic component (ref. 34-38), while the experimental data is in agreement with calculations of nonmagnetic defects in s_{\pm} gap structures (ref. 35). The dominant nonmagnetic scattering potential is widely used in theoretical modeling of LiFeAs, which was discussed by many authors as a reasonable approximation (*e.g.* ref. 38).

In our manuscript, no substitution atoms were deliberately introduced. The impurities are spontaneously formed, and most likely are clusters of Li atoms (the bare surface is Li surface) that reduce the surface energy. Considering all of the information, it is reasonable to assume that the non-magnetic scattering potential dominates in our case.

#Question 4: The authors state that the relevant impurities are actually clusters/complex impurity arrangements (that on top of that are strongly coupled to the host compound). In

addition, at different parts of the manuscript they highlight the importance of the symmetry properties of such clusters for the stabilization of the proposed MZMs. Given that controlled impurity manipulation is a real space atom-by-atom process it seems extremely difficult to imagine that the necessary coherent movements of MZMs required for braiding can be carried out in this way.

#Reply:

We thank the reviewer for the opportunity to clarify the discussion of Majorana manipulation. In our proposal, large scale impurity manipulation is required to create vortices with MZMs only for the initial setup. We do not expect to physically move vortices around to perform braiding or other manipulation and thus we do not require further movement of the impurity clusters. We completely agree with the referee that such operation would be extremely difficult and maintaining coherence would be impossible.

However, in general the coherent movement of MZM is extremely difficult in any Majorana platform. This is one of the main reasons why the first proposal of braiding MZMs in real space (Nat. Phys. 7, 412 (2011)) was soon updated by other strategies. The most up-to-date method for Majorana braiding is through the Hilbert space that is spanned by MZMs. By sequentially controlling the tunneling coupling among MZMs, the braiding process can be realized (*e.g.* PRB 84, 094505 (2011); New J Phys. 14, 035019 (2012); PRB 94, 235446(2016); PRB 95, 235305 (2017)). These ideas can be employed in a Majorana lattice designed by impurity manipulation and have a good potential for implementing a braiding method. However, a specific design is beyond the scope of our manuscript. To make those points clear, we added those explanations as a new section in the Methods of revised manuscripts.

We would also like to point out that the symmetry requirement for creating MZMs is easy to be realized for impurities, as the lower symmetry is more favorable. Moreover, besides impurity manipulation, we also proposed other more controllable methods for managing MZMs, *i.e.* by electrostatic gating and applying pressure.

#Question 5: It is undoubtedly true that strain and electrostatic doping can break C4 symmetry / shift the average chemical potential. And indeed adding more and more atoms to a cluster of impurities might induce ZBCPs. However, even if these are MZMs, it is not clear at all how homogeneous lattice deformations or atom-by-atom growth of an impurity can be utilized in order to create movable Majorana patterns for braiding. To be clear, I do not doubt that these are good strategies for the targeted creation / destruction of the observed ZBCPs but have concerns describing these as controlling MZM in the sense of enabling braiding as stated in the last sentence of the discussion.

#Reply:

As we clarified in the reply for #Question 4, it is not necessary to realize moveable Majoranas to perform braiding. Targeted creation / destruction of the observed ZBCPs could change the tunneling coupling among MZMs in a well-designed manner. It has the potential to be used in braiding MZMs in the Hilbert space which is one of the up-to-date strategies, and has been proved to be equivalent to braiding MZMs in the real space.

#Question 6: OK

#Reply:

We thank the reviewer for the careful review.

New Questions raised by the reply:

#Question Q1:

In light of the new discussion including the characteristics of the ZBCP generating impurities the authors state that these are (exclusively?) clusters. This implies an extended rather than a point scattering potential as assumed in the BdG equations. In how far are the proposed theoretical models even applicable to the experimental situation? *i.e.* can the authors add a discussion of the relevant length scales?

#Reply:

We thank the referee for the opportunity to clarify the performed calculations. In the simulations performed in this work we used extended Gaussian function to model the scattering potential of an impurity cluster as described in the Methods section. This is possible as the calculation is done using lattice model and thus the potential can extend over many lattice sites. Therefore, there was no assumption of point-like scattering in the calculation. The σ parameter of the Gaussian is 15% of the coherence length, so for the impurity and vortex in question, the distance of 4σ (so $\pm 2\sigma$ from the impurity center) is about 2.2 nm. We can compare this to the diameter of the cluster as visualized in topography measurement in Fig. 3b, which is about 2.5 nm. This means that the spread of the scattering potential used is comparable to the spatial extent of the impurity cluster, allowing us to estimate the impact of the impurity on vortex state energies and wave functions more realistically. We added a discussion in related section of the Methods.

#Question Q2:

In the new sections the authors imply and emphasize that strong scattering centres are necessary to observe MZMs as they are not observed in weak impurities. ‘Strongness’ seems to be characterized by lower E_{IBS} and greater electron doping (*i.e.* higher chemical potential shift).

Given that 14% of *impV* showed ZBCPs and 9 ZBCPs are observed there should be of the order of 50 *impV* that do not show ZBCP/MZM ‘which is likely to be related to their weakness’.

Given that the paper is about the controlled stabilization of MZM in *impV*, I would expect a more quantitative separation / statistics on what makes these 14% of *impV* special - *i.e.* would it be possible to show a distribution of E_{IBS} or the chemical shift across all *impV* and demonstrate that the observed MZM are indeed occurring in a special subclass of *impV*?

Would it be also possible to clarify which data points in Fig. 6 correspond to ZBCP hosing *impVs* and which not, in order to demonstrate that these really separate into a ‘strong’ and ‘weak’ regime?

#Reply:

We thank the reviewer for the thorough analysis, which prompted us to gather additional experimental data to further support our claims on this admittedly complex issue.

As we mentioned in the manuscript, the “strongness” of the impurity for creating MZMs in an *impV* is characterized by three aspects: (1) providing sufficiently large electron doping; (2) providing sufficiently large lattice strain; (3) influencing a sufficiently large volume of the sample. A large electron doping is thus one of the aspects of a “strong” impurity.

To increase the statistical sample, we have measured additional 18 vortices. The percentage of vortices with MZM (14% in *impV* and 11% in all of the vortices) is determined by an unbiased measurement of vortices one-by-one in a large area (150 nm by 150 nm). We note that this method is the most reasonable way to infer the distribution of different types of vortices in a material. Generally, not every vortex in the field of view is measured in a real experiment, which makes the ratio meaningless as it is calculated by using the total number of vortices measured as the denominator. We display the detailed behavior of vortices in this area in Fig. RR1.

After applying a 2-T magnetic field perpendicular to the sample surface, we observed 18 vortices in this area (150 nm by 150 nm). By carefully measuring $dI/dV(V,r)$ the line-cuts across each of the vortex (similar measurement as shown in Fig. 3a-3c), we determined their types as listed in the tables of Fig. RR1. It could be shown clearly in this measurement that the probabilities of *freeVs*, Majorana *impVs* with ZBCP, and ordinary *impVs* without ZBCP are 17%, 11% and 72% respectively. In addition, we also extracted the μ -shift of each vortex in this area following the same methods as introduced in Fig. 5. The extracted values are also listed in tables of Fig. RR1.

Fig. RR1 Vortex statistics in a large area. Left-top panel: the topography of this area. The circles mark the positions of vortices. The green, yellow and white colors indicate *freeV*, Majorana *impV* and ordinary *impV* respectively. Right-top panel: the corresponding vortex mapping measured at zero-bias. Bottom tables: The parameters of each vortex. Note that the error bar of μ -shift is about 1 meV, similar to what is used in the main figures.

In total, including the previously measured vortices, we obtained the μ -shift for 18 ordinary *impVs* (the case without ZBCPs) (13 cases listed in Fig. RR1 & 5 cases listed in Supplementary Table 2). The shape and symmetry of their pinning impurities can be well identified through topography measurement, which is accordant with “simple impurity”. On

the other hand, 11 Majorana *impVs* were measured (the case with ZBCPs) (2 cases listed in Fig. RR1 & 9 cases listed in Supplementary Table 1), and the μ -shift measurements were performed on 6 of them. The impurities for these vortices are more complex, either being a large cluster or a crowding group of simple impurities, as we mentioned in the Methods of the manuscript.

Following the suggestion of the reviewer, we prepared the μ -shift histogram for those two groups separately (Fig. RR2). It demonstrates that the Majorana *impVs* (pinned by stronger impurities) have larger μ -shift values. Although, as mentioned at the beginning of this reply, the “strongness” of the impurities is influenced by multiple issues simultaneously, the results shown in Fig. RR2 based on the single issue of μ -shift are fully consistent with the expectations, and it indicates the observed MZM occurring in a special subclass of *impV*.

Fig. RR2 μ -shift histogram statistics. For ordinary *impVs* (upper panel) and Majorana *impVs* (lower panel).

In Fig. RR3, we replot Fig. 6a with data points belonging to Majorana *impVs* (red) and ordinary *impVs* (blue) in different colors. The two types are roughly separated into two regimes, but rather strongly scattered. Again, this scattered behavior can be reasonably understood because the creation of MZM in an *impV* is not determined by a single parameter of μ -shift. To be clear, a good example to demonstrate this “multiple issues effect” is to check the details of the data point enclosed by a black circle in Fig. RR3. We note that this data point belongs to the *impV* #NM4 (see Supplementary Figure 3c & Supplementary Table 2). Although strong scattering potential of #NM4 induces small E_{IBS} and large μ -shift, the impurity preserves C_4 symmetry that violates one of the necessary conditions for MZM emergence, as we discussed in Fig.4. In addition, two pairs of helical Majorana modes may occur in the case of #NM4. As shown in Fig. 4d and the discussion in the section of Methods entitled “The case of non-ZBCP for weak impurities”, it has the featureless density of states and the *impV* appears to be similar to that in a *freeV*.

Fig. RR3 Replot of Fig. 6. The data points in red and blue color belong to Majorana or ordinary *impV*s respectively. The data point enclosed by a black circle belongs to *impV* #NM4 which preserves the C_4 symmetry.

The main purpose of Fig. 6a is to show general correlation between the ability of impurity to induce local doping of the sample and the E_{IBS} . The MZM appearance or absence is not the main issue we focused on that figure. Therefore, we would like to keep the appearance of Fig.6 the same to avoid too many details that would distract the readers. Moreover, this reply (also a public document following the policy of *Nat. Commun.*) could provide more information for the readers who want to know more details.

Overall, I have two remaining key concerns:

(i) It remains difficult to see how the proposed methodology for creation/annihilation of ZBCPs is sufficient to claim that braiding is possible in this system.

#Reply:

We provided explanations to this concern in the replies for #Question 4 and #Question 5. It is not necessary to exchange the MZM in the real space for the realization of non-Abelian braiding. The methodology for creation/annihilation of ZBCPs is in principle possible in the new proposals for braiding MZM in the Hilbert space (with those modes fixed in the real space).

(ii) The provided evidence that strength is the key driving parameter for the creation of ZBCP seems insufficient, as it is not clear from the data presented that the remaining 86% of *impV* not showing a ZBCP are indeed significantly weaker. This is key to the manuscript as the capability of controlling the occurrence of ZBCP as a function of this strength is one of the major conclusions.

#Reply:

We discussed this concern in the reply for #Question Q2. Following the constructive suggestion of the reviewer, we provided more data and analysis and made μ -shift histogram statistics on Majorana and ordinary *impV*s separately. The observed MZM are occurring in a distinguishable special subclass of *impV*. The native impurities investigated in this work promise Majorana tunability in LiFeAs and could also inspire other controllable strategies which can be explored in the future.

These two points relate directly to the key claims of the paper. To be clear, the experimental data in itself reporting the observation of ZBCPs at complex *impV* sites seems impeccable and their properties consistent with reports on the existence of ZBCP that are possible realizations of MZMs in related compounds. However, the authors themselves state in abstract and discussion that the key scientific advance of this paper is not the observation of these experimental signatures in themselves but rather that they occur in homogenous LiFeAs as a potential future platform for manipulating/braiding MZMs. Beyond the host system for these phenomena in this case being homogeneous, I feel not sufficient evidence is provided that with existing atom-by-atom manipulation a controlled creation of a ZBCP hosting cluster is reasonably feasible in the sense that (i) the necessary composition/structure/key characteristics of such clusters has not been sufficiently determined in order to distinguish them from non-ZBCP-hosting clusters and (ii) the controlled assembly is within the realm of existing atom manipulation techniques.

#Reply:

We appreciate the high evaluation of our observations by the reviewer and we are grateful for the careful consideration of the feasibility of future controllable manipulation schemes. However, we would like to point out that our experimental observations themselves are one of the key advances of this paper, which solves a long-term puzzle of LiFeAs, reveals new

Majorana physics related to bulk Dirac fermions, and promises a better Majorana platform of LiFeAs. We added some descriptions in Discussion section, in order to make this point clear.

As mentioned in previous paragraphs, stimulated by those very constructive questions from the reviewer, we make our proposal more feasible in this round of response by (1) providing more detailed characteristics of clusters hosting ZBCP (reply for #Question Q2); (2) clearly explaining the braiding strategies that need no real-space movement (reply of #Question 4 & 5). In addition, we also achieved atom manipulation in Fe(Te,Se) (Supplementary material of Nat. Commun. 12, 1348 (2021)), which can be also adapted to LiFeAs.

It is worth noting that using impurity is just one of the tuning methods for driving the vortex of LiFeAs into the topological regime. Vortex MZM induced by impurity tuning indicates the potential of LiFeAs itself. The physics revealed in this work implies that any tuning method, which could introduce sufficient electron doping and symmetry breaking effect in a large surrounding volume simultaneously, can create vortex MZMs in LiFeAs. Thus we provide an alternative and more controllable experiment design by combining electrostatic gating effect and piezoelectric stress with LiFeAs superconductor. In addition, in the past few months after submitting the current work, we found that external mechanical pressure can modify the chemical potential of LiFeAs by up to several tens of meV. We have finished ARPES measurements of the pressure-dependent band structure and will publish it as a separate work. The pressure device used in the ARPES measurements can be adopted for future STM experiments, and some experiments proposed in the current work for directly realizing tunability of vortex MZM on LiFeAs can be performed in the near future.

~~~~~

## To Reviewer #2

~~~~~

We greatly appreciate the careful review and constructive suggestions of Review #2, which helped us to strengthen this manuscript by including more experimental evidence and comprehensive discussion. In the following, we provide a reply for the remaining issues of the Reviewer #2.

Reviewer #2 (Remarks to the Author):

The authors have, in principle, addressed all my former comments and questions in a reasonable fashion. I therefore recommend publishing the revised manuscript after the authors have responded to one (minor) remaining issue:

In my earlier report, I asked the question (1-3 in the authors' response) "Could there be an impurity just below the surface, not visible in topography". This question is not answered. In fact, I'm surprised by the vigor with which the authors rule out such buried defects (specifically when working with only one value of the bias voltage). The authors should be very specific to what depth they think they see which type of impurity in their topography.

#Reply:

We agree with the reviewer that the topography measured using only one value of the bias voltage cannot rule out a buried defect just below the surface. However, the spectroscopic measurements of dI/dV spectra also do not indicate any position dependence along the analyzed linecut, which supports a clean case in this area.

In order to determine the conditions around the vortex shown in Fig. 2, we measured the zero-field dI/dV spectra at the same positions in which vortex bound states measurements were performed (Fig.RR4). Figs. RR4c & 4d show a hard superconducting gap with no impurity induced bound states or any other in-gap features that can be resolved from the spectroscopic measurements. Seeing that the zero-field superconductivity does not suffer any perturbations, we conclude that the vortex emerged in the same area is a free one, indicating that no impurity appears just below the surface in this specific case shown in Fig.2 and Fig. RR4.

Fig. RR4 Zero-field spectra in the area of a free vortex. (a) and (b). Zero-bias vortex mapping and topography of a *freeV*, respectively. (the same data shown in Fig. 2a and 2b) (c) and (d). Line-cut plot (a) and the corresponding dI/dV spectra (b) under zero field. The spatial positions for dI/dV measurements are indicated as the white-dashed line with arrow in (a) and (b). These positions are same as that in vortex line-cut measurements shown in Fig. 2c.

For the question about the depth of the impurity, we note that STM technique is incapable to make a specific and direct estimation about what depth of the electronic states can be detected. However, given that the tunneling spectra contain superconducting features which are Bogoliubov quasiparticle from the Fe-As layer underneath the bare surface, one can roughly infer that the detectable depth is at least 5 Å (or one unit-cell). Nevertheless, the STM measurement is not capable to identify the exact depth specifically for each impurity.

~~~~~

### To Reviewer #3

~~~~~

We greatly appreciate the careful review and constructive comments of Review #3, which have clearly strengthened this manuscript.

Reviewer #3 (Remarks to the Author):

In the revised manuscript, the authors resolved most of my concerns. In terms of novelty, I agree that although some related ideas were proposed before, an experimental realization is significant. In addition, the proposed "tunable Majorana device" could generate new thinking in the field of topological quantum computation. Therefore, it is my pleasure to recommend this work to be published in Nature Communications.

REVIEWERS' COMMENTS

Reviewer #1 (Remarks to the Author):

I would like to thank the authors for the careful consideration of my questions and the adjustments to the manuscript. I believe that my concerns were in principle addressed in the reply to a reasonable level, providing substantial additional information in their detailed answers to questions.

I therefore would like to recommend publication.

Reviewer #1 (Remarks to the Author):

I would like to thank the authors for the careful consideration of my questions and the adjustments to the manuscript. I believe that my concerns were in principle addressed in the reply to a reasonable level, providing substantial additional information in their detailed answers to questions.

I therefore would like to recommend publication.

#Reply:

We greatly appreciate the careful review and constructive comments of Reviewer#1 during the three-round review process, which have greatly improved this manuscript.